

# Dust size parameterization in RegCM4: Impact on aerosol burden and radiative forcing

Athanasios Tsikerdekis[1], Prodromos Zanis[1], Allison. L. Steiner[2], Fabien Solmon[3], Vassilis Amiridis[4], Eleni Marinou[4], Eleni Katragkou[1], Theodoros Karacostas[1], Gilles Foret[5]

[1]Department of Meteorology and Climatology, School of Geology, Aristotle University of Thessaloniki, 54124 Thessaloniki, Greece
[2]Atmospheric, Oceanic and Space Sciences, University of Michigan, Ann Arbor, MI 48109, USA
[3]Earth System Physics Section, The Abdus Salam International Centre for Theoretical Physic, 34100 Trieste, Italy
[4]Institute for Astronomy, Astrophysics, Space Application and Remote Sensing, National Observatory of Athens, 15236 Athens, Greece
[5]Laboratoire Inter-universitaire des Systèmes Atmosphériques (LISA), UMR7583, Universités Paris-Est Créteil et Paris Diderot, CNRS, Créteil, France

*Correspondence to*: Athanasios Tsikerdekis (tsike@geo.auth.gr)

**Abstract.** We investigate the sensitivity of two dust parameterizations of the regional climate model RegCM4 for the period 2007-2014 over the Sahara and the Mediterranean. We apply two discretization methods of the dust size distribution keeping the total mass constant: 1) the default RegCM4 4-bin approach, where the size range of each bin is calculated using an equal, logarithmic separation of the total size range of dust, using the diameter of dust particles and 2) a newly implemented 12-bin approach with each bin defined according to an isogradient method where the size ranges are dependent on the dry deposition velocity of dust particles. Increasing the number of transported dust size bins theoretically improves the representation of the physical properties of dust particles within the same size bin. Thus, more size bins minimize the error and improve the simulation of atmospheric processes. The radiative effects of dust over the area are discussed and evaluated with the CALIPSO Dust Optical Depth (DOD). This study is among the first studies evaluating the vertical profile of simulated dust with a pure dust product. Reanalysis winds from ERA-interim and the total precipitation flux from the observational gridded database CRU are used to evaluate and explain the discrepancies between model and observations. The new dust binning approach increases the dust column burden by 4% and 3% for fine and coarse particles respectively, which increases DOD by 10% over the desert and the Mediterranean. Consequently, negative shortwave RF is enhanced by more than 10% at the top of the atmosphere and by 1% to 5% on the surface. Positive longwave RF locally increases by more than $0.1 \text{W·m}^{-2}$ in a large portion of the Sahara desert, the northern part of the Arabian peninsula and the Middle East. The 4-bin isolog method is to some extent numerically efficient, nevertheless our work highlights that the simplified representation of the 4-bin approach underestimates the dust optical depth and the radiative forcing, a fact that should be taking into account by future researches that study the same region.



# 1 Introduction

Aerosols affect the energy budget of the planet through numerous feedback mechanisms. The direct aerosol climatic impact of -0.27W·m-2 and the indirect effect of -0.55W·m-2 was estimated in the most recent IPCC report, yet the forcing uncertainty remains the highest among all the other factors (Boucher et al., 2013). Aerosols strongly affect Earth's climate

by scattering and absorbing the incoming solar radiation (direct effect), burn low clouds through longwave heating (semi-direct effect) or by producing brighter clouds (first indirect effect) with longer cloud lifetime (second indirect effect) (Bangert et al., 2012; Hansen et al., 1997; Karydis et al., 2011; Lohmann and Feichter, 2001; Nabat et al., 2014; Ramanathan et al., 2001; Tegen, 2003). Small particles can effectively scatter/reflect the incoming shortwave radiation while large particles can more effectively absorb and re-emit on the longwave part of the spectrum (Liao and Seinfeld, 1998). Globally,

dust is considered to have a slightly negative radiative forcing -0.1W·m-2 (-0.3W·m-2 to 0.1W·m-2) on climate (Boucher et al., 2013). A recent detailed evaluation of RegCM4-CCM3 using satellite-based observations over the European domain highlighted that surface solar radiation bias can be attributed to an overestimation or underestimation of counteracting parameters, with cloud fractional cover, cloud optical thickness and aerosol optical depth being the most important (Alexandri et al., 2015). Since airborne dust play a fairly significant role on the total aerosol optical depth, it is important to

understand and improve the probable causes of dust optical depth biases.

Dust production begins when surface wind or wind at a certain height exceeds a threshold. In most studies, the threshold friction velocity is determined as the minimal wind velocity that is capable of sustaining particle movement (Iversen and White, 1982). Practical threshold wind velocity is depended on soil particle size, inter-particle cohesion forces and surface roughness features. Assuming a constant particle density, soil particle size affects their weight. Thus, large particles require

higher surface friction velocity to initiate dust movement. On the other hand the inter-particle forces (van der Waals, Capillary, Coulomb forces) acting upon small dust species, that depends on soil moisture and chemical composition, are higher compared to the larger particles and difficult to estimate (Shao and Lu, 2000). Due to the opposite dependency of particles weight and inter-particle cohesion forces size particles with diameter 60μm was estimated to have the minimum wind erosion threshold (Knippertz and Stuut, 2014). Particles of this size have a higher probability to be emitted first and

trigger the next stages of dust emission (saltation and disaggregation).

Mineral dust is produced mainly in deserts. Its production depends on surface wind, precipitation and changes in the vegetation cover (Tegen et al., 2000). The Sahara desert is the most important source of aeolian soil dust on the planet (Prospero et al., 2002). Dust from the Sahara is transported northward in Europe and Mediterranean on episodic events quite frequently (11.4 year-1) (Gkikas et al., 2013). The dust burden in the Mediterranean region differs spatially and seasonally

with the Eastern Mediterranean experiencing high dust load in spring and the Western Mediterranean reaching its peak in the summer and autumn (Moulin et al., 1998). The central Mediterranean can be considered a transition region where dust transport episodes can occur throughout spring to autumn (Israelevich et al., 2012). Depending on season and area of the Mediterranean, dust can originate from different places in the Sahara desert. Possible dust pathways have been identified for





each season using the TOMS AI satellite data (Israelevich, 2003). In spring for example, dust follows a long trajectory from the Bodélé depression to southern Algeria and then moves northward across the coast of northern Africa to reach Eastern Mediterranean. This long path affects the size of soil particles that reach the Mediterranean in spring in comparison with other seasons. Since dust particles have to travel for such a long distance, dry deposition through gravitational settling forces

coarse particles to deposit. Thus, the size of dust particles across Mediterranean in the spring (1.5µm) is half compared to the particles during summer and autumn (3µm), which follow a shorter trajectory (Israelevich, 2003).

The vertical dust distribution has remained uncertain in the past decade, because satellites measurement provided mainly the columnar aerosol optical depth and the stationary observations were spatially sparse. Modelling studies have shown different vertical dust distribution between the Mediterranean, the Atlantic and the Sahara desert (Alpert et al., 2004), while after the

10 launch of the CALIPSO satellite in 2006, our understanding on aerosol vertical distribution increased rapidly (Winker et al., 2009). The particle size distribution of dust changes over height with finer particles reaching higher in the atmosphere and therefore having a higher probability to be transported further because wind speed is also increasing with height. Considering that the fine particles scatter and reflect efficiently the incoming solar radiation, we can grasp the importance of the vertical dust distribution on the shortwave radiative forcing.

An important component that affects the transport and the radiative properties of dust in climate modelling is the number of transport dust size bins. The number of dust bin and the size range resolved in global climate models varies between models, with most using 1-6 dust bins spanning a size range of 0.01-25µm (e.g. Kinne 2003; Huneeus et al. 2011 and reference there in). The greater number of dust size bins minimizes model error especially for particle dry deposition and thus allows to more accurately simulate both the atmospheric dust burden and the interaction with radiation (Foret et al., 2006; Menut et al.,

2007). However, the high computational requirements of global climate models demands as few as possible dust size aggregates and consequently a low size range of aerosol species. Regional climate models with smaller domains of interest usually simulate between 4 to 12 dust size bins (e.g. Alexandri et al., 2015; Basart et al., 2012; Giorgi et al., 2012; Nabat et al., 2012; Solmon et al., 2008; Spyrou et al., 2013; Zakey et al., 2006), while dust transport models have conducted simulation experiments with up to 40 dust bins (Menut et al., 2007).

Dust particle size measurements in the Sahara desert from the Fennec 2011 aircraft campaign show that the dust coarse mode volume median diameter ranges between 5.8 and 45.3 µm (Ryder et al., 2013). Moving away from the dust source regions the mean/median size of dust particles drops dramatically. In the Mediterranean the mean diameter of dust particles ranges between 2-30µm (Goudie and Middleton 2001 and reference therein). It is evident that particles with diameter >25µm possess a major role in the burden of dust close and in some cases away from the dust sources. Thus, Foret et al. (2006)

proposed that dust size for the transported bins should range from 0.09µm to 63µm, considering both the total number and the mass distribution of soil particles.

This study presents new dust features in RegCM4, highlights the sensitivity of dust emission and deposition processes with various meteorological schemes, and analyzes the effect of size bin distribution on dust burden and radiative forcing. In Section 2 we describe the regional climate model RegCM4 and the gridded observational data LIVAS, ERA-interim and



CRU. We discuss previous studies on dust size parameterization and explain how the present work expands these studies further. In Section 3 we evaluate the regional climate model RegCM4 using the dust product LIVAS, where high resolution vertical aerosol extinction coefficient measurements were made available and new techniques for the discrimination of pure dust were developed (Amiridis et al., 2013). We also highlight the importance of the partitioning method and the number of

dust size bins by comparing two simulations in terms of the dust column burden and dust optical depth. Finally, we discuss the effect of the two binning methods on the dust radiative forcing both in the short and the longwave spectrum.

## 2 Material and Methods

### 2.1 RegCM4

The Regional Climate Model (RegCM) is a space limited numerical model developed at the National Center of Atmospheric

Research (NCAR) and the Abdus Salam International Center for Theoretical Physics (ICTP). Notably, it was the first area limited model used for long-term climate simulations (Giorgi and Anyah, 2012). Here we employ the RegCM4, which is described in detail in Giorgi et al. (2012). The hydrostatic core of the model, which is based on the fifth version of the Mesoscale Model (MM5) (Grell et al., 1994), restricts the minimum horizontal resolution of the model to 10km. Thus, the regional scale convective precipitation on RegCM4 is resolved through various convective scheme parameterizations. The

vertical computation of the atmosphere is applied on sigma levels. Land-Atmosphere interactions are analyzed with the Biosphere-Atmosphere Transfer Scheme (BATS) (Dickinson et al., 1993), while there is a recently implemented alternative option to use the Community Land Model (CLM4.5) (Oleson et al., 2013). The radiation transfer scheme used in RegCM4 is based on the NCAR Community Climate Model version 3 (CCM3) (Kiehl et al., 1996) with the additional option of the correlated-k Rapid Radiation Transfer Model (RRTM) (Mlawer and Clough, 1997; Mlawer et al., 1997) to improve radiation

processes in the longwave spectrum (Iacono et al., 2000).

The chemical part of the model contains gas phase chemistry (Shalaby et al., 2012; Steiner et al., 2014) as well as natural and anthropogenic aerosols (Solmon et al., 2006; Zakey et al., 2006, 2008). Most of the natural aerosols, such as dust and sea salt, are produced internally (online) in RegCM4 using the meteorological fields, while all the anthropogenic aerosols, organic carbon, black carbon and pollen require emission datasets.

The dust emission scheme is activated in a grid cell when the friction velocity, resolved as a function of RegCM4 simulated wind speed and surface roughness, is higher than the minimum friction velocity threshold. Dust saltation flux is calculated following Marticorena and Bergametti (1995) and Zakey et al. (2006), while the calculation of dust aerosol vertical flux from the saltation flux follows Laurent et al. (2008).

Particle size distribution (PSD) of the emitted dust and it's relation to surface wind conditions can be modelled using either

Alfaro and Gomes (2001) or Kok theory (2011a). The first theory, based on wind tunnel experiments, demonstrated that increasing wind speed close to the surface increases the portion of fine dust particles emitted (Alfaro et al., 1997). Based on these results (Alfaro and Gomes, 2001) suggested that the saltator kinetic energy increases with wind speed, thus the impact



produces more fine particles and affects the PSD of dust. On the other hand Kok (2011a) represented the impact of saltator on the surface to act as the fragmentation of brittle materials and suggested that the saltator impact speed does not depend on wind friction speed (Kok, 2011b). Using observational data he established a theoretical expression of particle size distribution independent of surface wind speed. Kok PSD theory shows a significant improvement of the simulated RegCM4

dust optical depth over the European and Northern African domain (Nabat et al., 2012), thus we use this parameterization in these simulations.

Surface roughness and soil moisture, which are essential for the calculation of threshold friction velocity and saltation fluxes, are provided by the surface scheme BATS (Zakey et al., 2006). The soil aggregate distribution, is used to determine threshold velocity and the saltation flux, differs from Zakey et al. (2006) and is taken from Menut et al. (2013), based on the

FAO texture class and spatial distribution. Emission processes take place only over desert or semi-desert grid cells. RegCM4 also accounts for possible subgrid partial desert cover emissions on grid cells dominated by different types of soil and soil texture.

Following the calculation of the dust mass emission fluxes, the tracer transport equation is applied for each transported bin (Solmon et al., 2006). The equation includes transport of the tracers through resolvable winds, horizontal and vertical

turbulent diffusion as well as vertical transport due to cumulus convection. The default number of dust size bins that are resolved for transportation in RegCM4 are four (Zakey et al., 2006) with the selection of each size bin defined according to the diameter of dust particles. The size range of each bin is calculated using an equal, logarithmic separation of the total size range of dust, using the diameter of dust particles. This method creates some biases that depend on how representative the selected size bins are compared to the actual dust particle size distribution in each bin. Following the methodology of (Foret

et al., 2006), we have implemented a new dust scheme that resolves twelve size bins instead of four. The specification of each bin is linked to the gradient of dry deposition velocity as a function of particle diameter. When the change rate of dry deposition velocity to particle size is high, more bins are created on that specified size range, while the opposite happens when the rate changes are slower. The improved partitioning of dust size bins as well as the higher number of bins theoretically enhances the representation of physical removal processes during transport, the estimation of the effective

radius of dust particles and the calculation of dust optical properties. Each transported bin is considered as a distinct tracer, which assumes that there is no mixing between the dust size bins.

Removal processes through dry deposition and wet deposition (washout) are included in the model. Dry deposition includes gravitational settling (a function of, particle size and density), brownian diffusion (acts mainly on small particles close to the ground) and turbulent transfer as well as impaction, interception and particle rebound (Zhang, 2001). Wet deposition

processes include a size dependent washout scavenging parameterization (Gong, 2003; Seinfeld and Pandis, 1998) and occurs for a small fraction of dust (10%) considered as soluble.

The optical properties of dust are pre-calculated for every size bin and each spectral band of the radiation scheme in use (CCM3 or RRTM) using Mie theory. Due to the high non-linearity of the Mie calculation, we calculated optical properties for each bin and each spectral band, based on a weighted average on size and wavelength assuming that the sub-size



distribution follows Kok (2011a), consistent with the emission size distribution. We ensure that the 12 and 4 bins options cover the exact same total size range and that the integrated optical properties over the whole distribution are conserved regardless of the number of bins. Figure 1 depicts the dust specific extinction coefficient, single scattering albedo and asymmetry parameter for both binning options. The differences for all the optical parameters are relatively small, because the calculations were performed for multiple effective particle radii within the range of each size bin and averaged in the end, instead of using the mean effective radius of each size bin. Using this method the optical properties between the two experiments are almost identical.

In our simulation, we use the default radiation scheme CCM3 to retain the same meteorological fields between the two binning experiments. In the current version of RegCM4, the RRTM scheme calculates cloud-radiation interaction using a Monte Carlo Integration of the Independent Column Approximation (McICA) (Pincus, 2003). The McICA method, although it provides a more detailed and complicated perspective on the horizontal and vertical radiative structure of the clouds, it introduces a random generated noise even when simulating the exact same experiment. In his work, Pincus (2003) accurately highlights that this random generated noise can be significantly reduced to zero with enough ensemble members. But sensitivity tests with RegCM4 show that the changes in the radiative effect between the two dust bin partitioning methods are already small, thus the results can be significantly altered even with the small perturbation contained in an ensemble mean. A downside of the CCM3 scheme is the simplistic representation of the longwave spectrum as far as the dust particles. In Section 3.3 we discuss the relevant impact of the binning methods in the longwave radiative spectrum between the two radiative transfer schemes.

## 2.2 LIVAS

The "LIdar climatology of Vertical Aerosol Structure for space-based lidar simulation studies" (LIVAS; Amiridis et al., 2015), is a 3-dimensional global climatic dataset derived from CALIPSO measurements and funded by the European Space Agency (ESA). CALIPSO obtains high resolution profiles of the attenuated backscatter of aerosols and clouds at 532nm and 1064nm, and retrieves aerosol optical properties below optically thin clouds, in clear skies and above clouds (Winker et al., 2009). Thus, in regions with frequent optically thick clouds, aerosol measurements are limited. In regions with an absence of clouds like the Sahara desert, measurements of aerosols species (most likely dust) are available more frequently.

LIVAS utilizes Level 2 (Version 3) product of CALIOP measurements (Amiridis et al., 2015). The CALIPSO Level 2 database determines the vertical location of cloud/aerosol layers (Vaughan et al., 2009), distinguishes the clouds from the aerosol layers (Liu et al., 2009), identifies aerosol layers into six subcategories (dust, marine, smoke, polluted dust, polluted continental and clean continental; Omar et al. 2009) and calculates the AOD for each selected layer (Young and Vaughan, 2009).

The LIVAS extinction dust product is corrected for the Lidar Ratio (LR) based on multi-year measurements performed by the ground-based lidar stations of the EARLINET lidar network (https://www.earlinet.org). The LR of dust particles depends on their refractive index and may vary for aerosols of the same type. The refractive index values rely upon the composition



of dust and most importantly on the relative proportion of clay-sized mineral illite in dust (Schuster et al., 2012). Thus, regions with different physiochemical dust characteristics leads to different LR values. The 0.3.1 version of LIVAS separates the globe into three regions, specified based on known dust sources and loadings with specific physio-chemical composition and LR for each region.

LIVAS has been evaluated against AERONET stations globally by Amiridis et al. (2015). The results show that the aerosol optical depth differences are between ±0.1 in most cases. Over the southwestern Sahara desert, LIVAS underestimates the AERONET AOD by -0.1, and this bias may be related with the dust underestimation of CALIPSO found in previous studies (Amiridis et al., 2013; Schuster et al., 2012; Tesche et al., 2013; Wandinger et al., 2010). Amiridis et al. (2013) showed that LIVAS correlates well with the Dark Target MODIS retrieval over sea, yet the correlation between MODIS Deep Blue and

LIVAS over Sahara are weak (results not shown). This could be attributed to the Deep Blue MODIS retrieval, which uses passive remote sensors for dust aerosol optical depth that take into account numerous assumptions (e.g. high reflectivity). Thus, LIVAS is a more reliable product over the deserts with higher accuracy than the products coming from passive remote sensing techniques. Furthermore, Georgoulias et al. (2016, submitted in ACP) showed that LIVAS correlates well with the high resolution TERRA MODIS and MACC dust optical depth over land covered regions in Eastern Mediterranean.

Monthly mean LIVAS data is available at a 1°x1° horizontal resolution with a vertical resolution ranging from 60 m (-0.5km and 21km) to 180 m (above 21km) (Amiridis et al., 2015). We concentrate on the troposphere where the vertical distribution of layers is constant. In our work we use the specialized LIVAS pure dust product which includes the extinction coefficient of pure dust calculated from the dust percentage of "dust" and "polluted dust" aerosol subcategories of CALIPSO (Amiridis et al., 2013). Since CALIPSO is a non-geostationary satellite, the obtained mean monthly profiles are highly dependent on the date and time the measurements were taken. Thus, a spatiotemporal mask was produced according to the exact flight

track of CALIPSO and applied to the nearest RegCM timestep prior to evaluation.

**2.3 Meteorological Data**

ERA-interim is a state-of-the-art global atmospheric reanalysis product developed by the European Centre for Medium-Range Weather Forecasts (ECMWF) (Dee et al., 2011). The data assimilation system uses various types of observational

data from ground-based stations, radiosondes, ships and satellites. Data availability starts from 1979 and it is extended forward in near-real time. Its assimilation scheme uses a 12 hour cycle, where observational data are combined with forecast information from the previous timestep in order to construct the global atmospheric conditions. ERA-interim reanalysis database is exceptionally useful over isolated areas like the Sahara desert, where meteorological measurements are limited. Wind velocity and direction, associated with dust emission and transport, were used to evaluate RegCM4 wind fields. ERA-

interim is used also as boundary conditions in several regional climate models (e.g. RegCM4) for hindcast simulations.

Climate Research Unit (CRU) data is a gridded global climate database of monthly meteorological measurement from ground-based stations (Harris et al., 2014). The dataset includes surface measurements of six meteorological variables, notably precipitation and temperature. Stations are interpolated into a 0.5x0.5 grid that covers all the land surface of the



planet (except Antarctica). Data availability peaks between the period 1950-2000 and drops dramatically in the last decade. During the period 2007-2014 and around the area of Sahara desert, more than 600 monthly measurements of precipitation are included in the database from various stations. However, we note that most of the continual measurements occur at stations close to the border of the desert, and station coverage is generally limited over the Sahara.

## 2.4 Dust particles size discretization

Dust tracers in climate models are usually represented by describing the size distribution with a specified number of bins defined according to the dust particle diameter (D). Increasing the number of transported dust size bins improves the representation of the physical properties and behavior of dust particles within same bin. However, the computational cost, especially in climate studies, generally limits the number of bins used to describe (dust). The transported size bins must be separated in an affordable number of bins and with a method that minimizes the numerical inaccuracies due to dust bin partitioning.

The isolog method, which partitions the dust bins in equal range of log D, is frequently used to specify the range of the dust size bins in climate models (Huneeus et al., 2011 and reference therein). An alternative partitioning method was introduced in (Foret et al., 2006), where they divide the total size range in each bin according to the changes of dry deposition velocity with size. Using a simple one-dimensional box model they simulated an experiment with a detailed particles size distribution that used 1000 size bins within the range of 0.001-100μm. They used this experiment as a reference to evaluate the differences between the isolog and isogradient method. The error ratio between the total dust particle mass (or number) simulated with the given particle size bin scheme (isolog or isogradient) and bin number (4-30bins), to the total dust particle mass (or number) simulated with the reference size distribution (1000bins), showed that for a given number of bins, isogradient scheme scored much better in terms of total dust mass, but isolog scheme produced somewhat better results in term of total dust particle number. The isogradient method yielded errors lower than 2% with regard to the total particle number and the representation of aerosol optical depth when using more than 8 dust bins (Foret et al., 2006).

Following the above results Menut et al. 2007 implemented the isogradient scheme into the 4D transport model CHIMERE-DUST and did several nested simulations over the Sahara region. To exclude any biases introduced by the emission, deposition and dynamics of the model they evaluated their results using a reference simulation that resolved 40 size bins. They concluded that the isogradient scheme reduced the errors by a factor of 2 compared to the isolog method and that 6 size bins are not sufficient to reproduce the correct concentration levels of dust even with the methodology proposed by Foret et al. 2006.

Both of these studies have shown that using a higher number of bins and the use of isogradient scheme improves the validity of the simulated results. However, they evaluate only the effect of dust bin partitioning on the size distribution dust emissions, either with a simple one-box model (Foret et al., 2006) or a three-dimensional dust transport model CHIMERE-DUST (Menut et al., 2007). In this manuscript, we take this research further by comparing the method and number of dust



size bins with in RegCM4 results, and evaluate using observational data, account for biases introduced by the balance of production and loss processes, the calculate of optical properties of dust and evaluate these results with satellite products.

## 2.5 Experimental set-up

The key parameters of the RegCM4 simulations are presented in the Table 1. The simulation domain includes the Sahara desert and the largest part of the Arabian Peninsula (Figure 2), which captures the two main sources of dust in the Earth (Tegen, 2003). To reduce the contribution of dust from outside the domain requiring chemical lateral boundary conditions, the domain extends south of the Sahara desert. Two 8-yearly simulations (4 dust size bins and 12 dust size buns) were performed from September 2006 to November 2014, excluding in our analysis the initial 3 months as a spin-up time. The 4 dust bin experiment (DUST4) implements the isolog approach for the partitioning of the dust size bins, while the 12 dust bin experiment (DUST12) uses the isogradient method. Both experiments use the Kok (2011a) dust particle size distribution theory.

To ensure that the two experiments are identical in every way except the dust bin number and partitioning method, we have both distributions span the exact same total size range and conserve the integrated optical properties (over the whole distribution) whether we consider 4 or 12 bins. Furthermore, there is no interaction between the dust particles and the radiation fields (e.g. no direct aerosol feedback on climate) to ensure that there are no meteorologically driven changes between the two simulations. The radiative forcing of dust is calculated using separate radiation call during the simulation. Consequently, the meteorology of the two experiments is identical.

The initial and boundary conditions of the mean monthly dust burden were acquired from a global simulation (CAM + EC-EARTH), for the decades 2000-2009 and 2010-2019. Therefore, a constant static map was used as boundary conditions for each month. A certain degree of error is expected at the boundaries of our domain that will not capture individual dust events or strong interannual variability. However, our analysis shows that the boundary forcing in the examined sub-regions is negligible in the dust annual cycle column burden of fine and coarse particles (Section 3.1). Because the aerosol lateral boundary conditions used a 4-bin dust model, we modified the boundary conditions for the 12 bin experiment. The new bins were defined according to the ratio between the 4 and 12 bin size range and size limits. Multiplying this ratio by each month and atmospheric level we have created the necessary global boundary condition for the 12 bin dust scheme. This method assumes that the number of particles is homogeneously distributed in each bin.

The dust size range of the boundary conditions dataset spans 0.01µm-20µm. Therefore, coarse dust size bins in both DUST4 and DUST12 scheme (e.g. that exceed the upper limit of the 20µm) are not influenced by boundary conditions. This does not introduce a significant bias in our results since large particles (> 20µm) have a short atmospheric lifetime and the main dust sources from outside the domain are located far from our simulation boundaries.





**Table 1. Simulation options implemented in RegCM4.**

| | |
|---|---|
| Grid Dimensions | 140 x 160, 18 sigma levels |
| Horizontal Resolution | 50km |
| Meteorological Boundary Conditions | ERA-Interim (Dee et al., 2011) |
| Surface Model | BATS (Dickinson et al., 1993) |
| Chemical Boundary Conditions | CAM + EC-EARTH |
| Cumulus Convection Scheme | Tiedtke (Tiedtke, 1989) |
| Transfer Radiation Scheme | CCM3 (Kiehl et al., 1996) |
| Moisture Scheme | SUBEX (Pal et al., 2000) |
| Planetary Boundary Layer Scheme | Modified Holtslag (Holtslag et al., 1990) |
| Dust Tracers | DUST4 (4bins, isolog) |
| | DUST12 (12bins, isogradient; Foret et al., 2006) |
| Dust Size Particle Distribution | Kok (Kok, 2011a) |

## 3 Results

### 3.1 Evaluation

The 4 bin simulation (DUST4) was evaluated against the climatological LIVAS Dust Optical Depth (DOD) (Figure 3). The

5   high dust belt (DOD > 0.2) between 15°-25°N in RegCM4 generally matches the observed DOD of LIVAS with some spatial inconsistencies. We discuss these differences for key regions (Figure 2) according to the mean bias, the lower (LCI) and upper (UCI) 95% confidence intervals of the mean and the percent bias (Pbias). RegCM4 underestimates DOD in a zone south of the Sahel around to 10°N by about 0.05. Over the eastern Sahara (ESah), RegCM4 overestimates the climatological DOD by 0.083 (LCI:0.078, UCI:0.088, PBias:63.1%), mainly in Chad and expanding to Libya, Egypt and northern Sudan. In

10  western Sahara (WSah) where the DOD is higher, the model mean overestimation is 0.043 (LCI:0.039, UCI:0.047, PBias:26.2%). Over the Mediterranean all the grid points show a weak overestimation of DOD which for the three Mediterranean regions combined is 0.027 (LCI:0.026, UCI:0.029, PBias:54.7%).

Figure 4 depicts the seasonal cycle of DOD in LIVAS observations and the DUST4 simulation. In the Mediterranean the observed monthly mean DOD values are typically less than 0.1, with the RegCM4 maximum values reaching 0.15 in some

15  months (Figure 4a-c). Although there is an almost constant overestimation of DOD for all Mediterranean regions, the observed annual cycle maximum and minimum values are simulated adequately and we note that generally, the monthly DOD values are relatively low. In the western Mediterranean DOD peaks in summer (Figure 4a) while the eastern Mediterranean peak is in spring (Figure 4c). Generally, the RegCM4 DUST4 simulation overestimates DOD with a secondary annual maximum in spring and summer over the western and eastern Mediterranean, respectively. The central



Mediterranean is the transition area that receives dust throughout the year from transport paths that affect both Eastern and Western Mediterranean (Israelevich et al., 2012). Therefore, it exhibits a broad maximum that peaks mainly in spring and remains until late summer (Figure 4b).

Over desert and semi-desert areas, DOD values are higher and show greater intra-annual variability and amplitude in comparison to the Mediterranean subregions (Figure 4d-f). Western Sahara DOD shows a strong summer maximum of 0.35 that is seven times higher than winter values (Figure 4d). RegCM4 simulates accurately the mean monthly values in the first six months and overestimates in the summer and autumn by 0.03-0.07. In the eastern Sahara, there is a significant overestimation of DOD during summer and autumn where the positive bias of the model is more than 0.1 in most months (Figure 4e). The Sahel annual cycle is affected by the southward transport of dust from Sahara and the seasonal movement of the Inter Tropical Convergence Zone (ITCZ) (Ridley et al., 2012; Rodríguez et al., 2015). The local emission sources are activated mainly during winter (dry season) when soil moisture is low but DOD peak in summer (Figure 4f), indicating that the dust annual cycle is strongly affected by the inflow of dust from the Sahara into the Sahel. RegCM4 underestimates DOD during August-January and overestimates it during March-June (Figure 4f).

To understand the simulated processes of dust production-destruction-transport in the model and explain the discrepancies identified in the DOD annual cycle, we investigate the annual cycle of dust column burden for coarse-silt size (Dd > 2.5μm, Figure 5) and fine-clay size (Dd < 2.5μm, Figure 6) dust particles alongside the production/destruction column tendencies of dust. Fine particles dominate the annual cycle of the total column burden and control its seasonal variability, whereas coarse particles display a weak seasonal variability and its intra-annual amplitude is generally negligible in comparison to fine particles. The major tendencies that controls the dust column burden in both fine and coarse particles is emission and sedimentation. Additionally, vertical turbulence, horizontal and vertical advection plays a considerable role in the desert regions for fine particles. The fine column burden annual cycle is anti-correlated with the absolute values of production/destruction tendencies in the desert, e.g., during the warm season the absolute values of tendencies tends to be small and fine dust column burden high, with the opposite in the cold season. In most regions, fine and coarse particles peak during the same month, enhancing the total column burden seasonal variability (e.g. western Sahara, Sahel; Figure 5e-f, Figure 6e-f). However, in the eastern Sahara (Figure 5d and Figure 6d), fine and coarse annual cycles follow a completely different pattern. Coarse particles peak in spring while fine particles exhibit a broad spring-summer maximum that peaks during late summer. This indicates that fine particles are responsible for the overestimation of DOD in the Eastern Sahara as their burden and the size specific extinction coefficient (Figure 1) is higher in comparison to the coarse dust particles.

Over the Eastern and Western Sahara both emission and sedimentation fluxes display a distinct spring maximum in March (Figure 5d,e and Figure 6d,e), that does not explain the annual cycles of dust column burden. The ratio of emission to dry deposition flux is connected to the dust column burden, rather than the absolute values of emission and dry deposition flux. Values greater than 1 demonstrate periods where the emission dominates in comparison to dry deposition rate, while values smaller than 1 depicts the dominance of deposition. Coarse particles, due to their low atmospheric lifetime, are deposited close to their source region and shortly after emission, therefore exhibiting a very low seasonal variability. Fine particles on





the other hand remain in the atmosphere for longer and can travel in long distances, affecting substantially the emission to dry deposition ratio.

In the Western Sahara the fine particle emission to deposition ratio for the full simulation period is 1.46, which indicates that the model emits 46% more fine particles compared to its deposition. The coarse particle ratio is close to 1. In Eastern Sahara the emission to deposition ratio is much higher than in the Western Sahara, with values of 1.80 and 1.31 for fine and coarse particles, respectively. The strong and almost constant annual NNE and NNW winds in the region transports dust away from the source region increasing the dust outflow of the Eastern Sahara (Figure 7). Thus, deposition in the Eastern Sahara decreases and the emission to deposition ratio rises. Furthermore, the fine particle emission to deposition ratio steadily increases from March to August, leading to an accumulation of fine dust particles in the atmosphere that increases the column burden and the DOD. In the Sahel throughout the year, emissions are two times greater than the deposition for fine particles, with a mean ratio 2.26. As discussed later, RegCM4 exhibits some biases in the total precipitation annual cycle that influences this ratio.

The column burden of dust can be affected by numerous meteorological variables that may alter the emission, deposition or the circulation and transport of dust. Thus, we have evaluated these variables with the observational gridded and reanalysis data to explain the spatial biases observed in the DOD. We chose the ERA-Interim for the evaluation of wind fields (surface, 925hPa and 850hPa) in order to have large spatial coverage and a long-term, continuous availability of data. The surface wind velocity in RegCM, that constitutes the main driver of dust production through erosion, is comparable to ERA-Interim (FigureS 1). We observe some positive and negative biases over the desert that generally are less than 1m·s-1. Although the mean wind velocity is linked with dust emission, wind gustiness and dust emission fluxes exhibit higher temporal and spatial correlation (Engelstaedter and Washington, 2007). A wind gust is defined as a sudden increase of wind speed that last less than 20 seconds. In our simulations, wind speed is calculated within the internal timestep (120 seconds) of our model and wind gust is considered as the maximum value of wind velocity within the last output timestep (6 hours). Typically, this calculation method is applied in most climate models and reanalysis products with different internal and output timesteps. Therefore, it is difficult to draw conclusions when comparing the wind gust in RegCM and ERA-interim, because the wind gust was calculated in different time frames. Alternatively we calculated the mean field of the 6hourly timesteps that exceeded the 0.9 percentile value of wind velocity, averaged over the three desert subregions (Eastern Sahara, Western Sahara, Sahel) (FigureS 2). The 0.9 percentile value of the averaged wind over the desert is 4.93m·s-1 and 5.92m·s-1 for ERA-interim and RegCM respectively. RegCM clearly overestimates locally high wind velocity over the desert by 1-3m·s-1, which indicates that the model potentially overestimates dust emission flux.

In Egypt and northern Sudan, the strong surface NNE winds depicted in ERA-interim is underestimated by RegCM, which may lower the simulated production of dust in the Eastern Sahara. But considering that the same wind pattern continues at both 925hPa and 850hPa and that the larger amount of dust is concentrated on these layers (Figure 7), it can also decrease its southward transport and the outflow of dust in Eastern Sahara. In Figure 8, the annual cycle of the meridional wind component confirms that during the summer (June-September) the southward wind is underestimated by the model by more



than 1m·s-1. Thus, a high load of dust remains stationary for longer period over Eastern Sahara that likely increases the modelled column burden and the DOD (Figure 4e).

Away from the Sahara region and especially in semi-arid environments, like the Sahel, precipitation can affect both emission and wet deposition processes. According to CRU database RegCM overestimates precipitation in the Sahel by 10-20 mm·month-1 (FigureS 3) which increases wet deposition as well as soil moisture and vegetation (Engelstaedter et al., 2006). Consequently, more dust is deposited and less is emitted. According to the annual cycle of total precipitation (Figure 9) the overestimation of precipitation during April, May and June probably contributes into the underestimation of DOD (Figure 4f). The same process prevents the southward transport of dust to lower latitudes and is partly responsible for the strong underestimation of DOD close to the Gulf of Guinea.

The vertical distribution of dust is evaluated for the six subregions for vertical tendencies (advection, convective transport, vertical turbulence and sedimentation) in Figure 10. The first 200m of Dust Extinction (DEX) values were excluded from the analysis due to measurement restriction of LIVAS. The tendencies are illustrated in the plot as percentages for the layers 0.2-5km and 5-10km. DEX is overestimated by RegCM in all subregions in the middle and upper Troposphere. More than 95% of dust is located in the first 5km from the surface according to DEX measurements from LIVAS. However, RegCM places more dust at higher altitudes, with only 80-90% located between 0 to 5km. In Western Sahara and Sahel, DEX is underestimated by RegCM until the height of 5km (Figure 10d and 10f) and above that height there is a constant positive bias that decreases with altitude. The large observed DEX values between 3km and 5km over the Sahel indicates the high amount of fine dust particles that reach higher altitudes as compared to the Eastern and Western Sahara. In the Eastern Sahara the DEX is overestimated between 3-13km. At the western, central and eastern Mediterranean regions, the DEX is overestimated above about 2-3km and underestimated below this height (Figure 10a-c).

In the model, the vertical transport of dust is controlled by four main processes: sedimentation, vertical turbulence, vertical advection and convective transport. The sedimentation tendency is always negative and equally important in all heights, whereas vertical turbulence plays a major role in the first couple kilometers of the atmosphere and vertical advection and convective transport, prevail in higher altitudes. As shown in Figure 10, in first 5km of the atmosphere sedimentation exhibit negative values (contribution >-40%), which removes dust from the atmosphere. On the other hand, vertical turbulence is the main force that raises dust upwards (contribution>50%). The planetary boundary layer (PBL) scheme based on Holtslag et al. (1990) produces high vertical turbulences in the first couple of kilometers in the atmosphere, and this may contribute to the underestimation or overestimation of the modelled DEX profile as compared to LIVAS measurements. We have simulated 2008 with the alternative PBL scheme option in RegCM4 based on Bretherton et al. (2004), which suggests an improvement as far as the vertical distribution of dust in the PBL, although column DOD and DEX bias increases, especially over Eastern Sahara (FigureS 4). The vertical advection and more importantly convective transport between 5 and 10km show positive values on average, which transports dust upward. Over the Mediterranean region vertical advection (>20%) and convective transport (>30%) contribution is equally important (Figure 10a-c), while in Sahara desert and Sahel convective transport is the most significant factor (>65%)(Figure 10d-f). Considering the overestimated DEX profile in this





altitude range it suggests that the cumulus convection activity/convective transport mechanism is overactive in Tiedtke scheme (Tiedtke, 1989) or another negative sign process (e.g. sedimentation) is not properly represented by the model. Discrepancies in the vertical distribution of dust can also be misinterpreted from possible local emission errors, the lack of simulated vertical levels or wet deposition biases. Further research is needed in this regard to reduce the mean vertical
distribution bias of dust in the model.

**3.2 Comparison of 4-bin and 12-bin experiments**

The optical properties for the 4 and 12 dust size bins simulations were calculated using a Mie scattering code for each CCM3 radiation band. As noted above, the meteorological forcing between the two models is the same, therefore the emission fluxes which depends on the model's resolvable winds remains the same. Thus, the changes in column burden between the
two experiments can only emerge from the new dust size discretization that theoretically improves dust transport and dry deposition processes, and the changes in DOD can only be attributed in changes related to transport and deposition.
Figure 11 compares the DOD coarse and fine column burden between the DUST4 and DUST12 for the full simulation period. The DOD percent increase is between 10.4% and 13% for all the subregions. Furthermore, there is a distinctive increase by 0.04 with the 12-bin model over the Sahara desert and especially along the Sahel region where the DOD values
are higher (Figure 11c). In comparison with the DUST4 simulation the DUST12 simulation increases the deposition lifetime (column burden/total deposition flux) by 3.5 hours and 2 minutes for fine and coarse particles respectively. Consequently, that increases the dust column burden of fine (+4%) and coarse (+3%) particles (Figure 11f,i). The changes in the fine particles correlates better with the changes in DOD, because dust extinction coefficient is much higher for fine particles (<2.5μm) (Figure 1). Over the Middle East and the northern part of the Arabian Peninsula we observe a distinct increase on
the coarse dust column burden by 10mg·m-2.
The dust bin resolution also depends on the emission size distribution considered for the dust. In our study we use the Kok (2011a) dust Particles Size Distribution (PSD), which as already mentioned improves the DOD and the seasonality over the Sahara region (Nabat et al., 2012). But Kok (2011a) PSD drops very fast for large and small dust particles. Other typical dust size distributions (e.g. Alfaro et al. 1998; Zender et al. 2003) do not show such a sharp drop when increasing or decreasing
the dust particle diameter (Kok, 2011a), thus they could be much more sensitivity to the binning partitioning method and number which will generate bigger changes in dust column burden and DOD.
Figure 12 shows the annual cycle of DOD for the DUST4 and DUST12 experiments. The monthly differences of the two experiments against LIVAS can be found in the supplementary FigureS 5. In all regions, the DUST12 experiment increases the DOD in comparison to DUST4. This is attributed to the increase of lifetime in fine and coarse dust particles in DUST12
with respect to DUST4 as also discussed earlier. Previous studies (Foret et al., 2006; Menut et al., 2007) revealed that the new approach in dust size bin partitioning and number (which is also adopted in our DUST12 experiment) more realistically simulates the transport and dry deposition of the dust size bins. This does not imply that the simulated biases will be reduced for all regions in the DUST12 experiment. The major factor that controls and potentially creates the first order biases in dust



models is the balance between the emission and sedimentations terms (e.g. Figure 5, Figure 6). As discussed in Section 3.1, some positive DOD biases (e.g. Eastern Sahara) might be due to an underestimation of the outflow of dust or a local overestimation of the emission flux from surface wind velocity errors. Therefore, although the new dust size parameterization theoretically improves dry deposition, it does not necessarily regulate or improve the biases that originate
from other processes.

## 3.3 Radiative Forcing

Dust particles can interact both with the shortwave and the longwave radiation, creating a dimming or a heating radiative effect on climate (Liao and Seinfeld, 1998). Some of these processes are illustrated for the shortwave spectrum in FigureS 6. According to the model the radiation forcing over the Sahel and the North Atlantic ocean is great than -5W·m-2 (Figure
13a,b). The albedo on the desert areas is already very high and in most cases, surface albedo does not change from the suspended dust. However, close to the high emission dust source of Bodélé depression we can observe positive RF values. In the model dust aloft decreases the already high albedo of the desert and creates a positive RF at the TOA. In the Bodélé depression, sediments were deposited during the Holocene in the bed of the Megachad paleolake and the large diatomite sediments formed there have high albedo values that are visible in satellite images (Bristow et al., 2010). It is interesting that
RegCM4 simulates a positive radiative response caused by the combination of high surface albedo values and high emission fluxes on that area, and the change to DUST12 decreases this positive response (Figure 13c). The DUST12 experiment enhances the negative radiative forcing in the central and eastern Mediterranean by -0.24W·m-2 (10.5%) and in eastern by -0.18W·m-2 (8.7%). The highest absolute changes are located over the Sahel where the negative radiative forcing strengthens by -0.41W·m-2 (12.1%).
The radiative process that backscatters and reflects the incoming solar radiation is the dominant global mechanisms that affects climate. As shown in Figure 13d and Figure 13e, dust prevents more than 20W·m-2 of incoming solar radiation from reaching the surface on the desert and 5-10W·m-2 on the Mediterranean. The highest absolute differences between the two experiments are located on the western part of Africa between 15º-20ºN, where the highest fine dust burden differences between DUST4 and DUST12 were noted (Figure 11i). These negative changes are located in the region with the most
persistent dust concentrations (Figure 11g,h) and changes in the dust size bins might have a bigger effect over the long range Trans-Atlantic transport than local forcing. It is worth discussing the source of the surface positive radiative forcing differences between the DUST12-DUST4 experiment (Figure 13f), which is related to the TOA discussion. The positive RF at the TOA over the Bodélé depression is reduced in the DUST12 experiment, allowing less downward shortwave radiation to reach the surface. Consequently, there is less available downward radiation to be scattered and reflected on these areas,
and the negative radiative forcing in DUST12 is smaller than DUST4. As a result the difference (DUST12-DUST4) is positive. The simulated shortwave radiative forcing by the 4-bin isolog method is to some extent numerically efficient and acceptable. Nevertheless, our work emphasize that the simplified representation of the 4-bin approach underestimates the



direct radiative forcing of dust by 13.7% (-0.29 W·m-2) and the SW SRF radiative forcing by 1.8% (-0.23 W·m-2). A fact that should be taking into account by future researches that study the same region.

On the longwave spectrum, the RF is always positive. Coarse dust particles in the atmosphere absorb the upward longwave Earth's radiation and remit it either towards the surface or upwards to space. A part of the Earth's longwave radiation is absorbed by dust and limits the portion of upward longwave radiation that reaches the TOA, making the RF positive (Figure 14a,b). A fraction of the absorbed longwave radiation is remitted back towards the Earth and increases the downward longwave radiation on the surface, creating a positive RF (Figure 14d,e). In both cases the DUST12 experiment enhances the positive RF by 0.1W·m-2 in a large portion over the Sahara desert, the northern part of the Arabian peninsula and the middle east (Figure 14c,f). More specifically at the TOA the positive radiative forcing enhances over the eastern and western Sahara by 0.08W·m-2 (6.9%) and 0.07W·m-2 (5.9%) while in western, central and eastern Mediterranean increases by 0.02W·m-2 (7.8%), 0.03W·m-2 (7.8%) and 0.04W·m-2 (8.3%) (Figure 14c). Similarly, at the surface the positive radiative forcing rise by 0.08W·m-2 (3.0%) and 0.09W·m-2 (2.7%) in eastern and western Sahara and 0.9W·m-2 (6.3%), 0.08W·m-2 (4.8%) and 0.9W·m-2 (6.5%) in western, central and eastern Mediterranean (Figure 14f).

However, the current treatment of the optical properties of dust in the longwave spectrum of CCM3 scheme is limited and does not account for specific absorption coefficient for each dust size bin. Furthermore, the CCM3 longwave bands are concentrated on the absorption of $H_2O$ and $CO_2$ and they do not integrate in detail the absorbing aerosol component part of the longwave. Therefore, we have conducted two similar dust experiments for June 2008 using the radiation transfer scheme RRTM, which is known for its detailed longwave calculation. We note again that the current version of RRTM (+McICA) produces a random generated noise on radiation fields, therefore the dust emission fluxes are not identical between the DUST4 and DUST12 experiments. Our preliminary simulations using the RRTM scheme (FigureS 8), have shown notably lower longwave dust RF over the desert in comparison to CCM3 (FigureS 7) for the same period and roughly the same DOD levels and spatial patterns. More specifically, the DOD and the longwave radiative forcing averaged above the Sahara for the CCM3_DUST4 experiment is DOD=0.31, SRF=3.5W·m-2 and TOA=2.1W·m-2. While the related values for the RRTM_DUST4 experiment is DOD=0.29, SRF=1.9W·m-2 and TOA=0.6W·m-2. Although DOD increases in the 12 bin experiment in both radiation schemes (FigureS 7c, FigureS 8c), the longwave RF changes display a striking difference. With the CCM3 radiation scheme, the longwave radiation linearly increases with higher DOD and dust burden values (FigureS 7f and i). In comparison, the RRTM scheme uses specific absorption coefficient for each dust size bin, taking into account the fine/coarse dust burden changes. Thus is exhibits local increases or decreases of the longwave RF (FigureS 8f and i), according to the changes of fine to coarse dust burden. Overall the total longwave RF increase in RRTM is smaller compared to CCM3 when spatially averaged over the Sahara desert.





## 4 Conclusions

In the present study, we investigate the role of the modelled particle size distribution on the emission, total column burden and radiative effects of dust in a regional climate model. We evaluate the regional climate model RegCM4 dust optical depth and dust extinction with the LIVAS dust product to understand potential model biases and link these with size-dependent

emission and atmospheric processes. Generally, RegCM4 overestimates dust optical depth over important source regions such as the eastern and western Sahara by 0.083 (Pbias:63.1%) and 0.043 (PBias:26.2%) respectively and in Mediterranean by 0.027 (PBias:54.7%).

The dust optical depth annual cycle of LIVAS correlates with RegCM4 adequately in most regions. In the western Mediterranean, RegCM shows an annual cycle with a summer maximum similarly to LIVAS and in eastern Mediterranean,

RegCM captures the spring maximum observed in LIVAS but additionally it illustrates a secondary maximum in summer. In the western Sahara, the RegCM DOD annual cycle correlates with LIVAS, although there is a constant overestimation during summer and autumn while in the eastern Sahara RegCM shows a broad spring-summer maximum while the observations show a clear spring maximum. In the Sahel, the model captures the observed DOD summer maximum.

We then evaluate the modelled emission and sedimentation terms, as these are the most important factors for the production

and deposition of dust. Additionally, vertical turbulence, horizontal and vertical advection pose a considerable negative tendency for the fine dust particles (Dd < 2.5μm). According to the annual cycle of the meridional wind component in ERA-interim, the model underestimates the southward wind in eastern Sahara and probably decreases the outflow of dust in the area. Furthermore, the 0.9 percentile value of the averaged wind over the desert is 4.93m·s-1 and 5.92m·s-1 for RegCM and ERA-interim respectively, which indicates that the model potentially overestimates dust emission flux. At the Sahel during

April, May and June the total precipitation is overestimated by the model in comparison with the CRU database. Thus, wet deposition is enhanced and emission flux is decreased since the ground has higher moisture. Overall, this process-level analysis suggests that meteorological drivers such as wind and precipitation may explain some model biases with observations.

These changes in the physical processes then link to the radiative properties that affect climate. We evaluate the Dust

Extinction (DEX) from the model with LIVAS observations, and find the RegCM overestimates DEX in all subregions in the middle and upper troposphere, that can be either attributed to an overestimation of dust convective transport or to an underestimation of some other deposition process (e.g. sedimentation). In the Western Sahara and Sahel, DEX is underestimated by RegCM below 5km, suggesting that the emissions and deposition processes are not well-balanced in the model. Our one year sensitivity study showed that the PBL mixing scheme (e.g., changing from Holtslag et al. (1990) to

Bretherton et al. (2004)) can potentially improve the dust vertical distribution in the boundary layer by reducing overactive mixing.

The effect of the two different approaches on the number and the partitioning method of dust size bins was investigated in the model with two experiments: a 4 size bin simulation based on the isolog approach (DUST4) and a 12 size bin simulation





based on the isogradient approach (DUST12). The DUST12 experiment increases the deposition lifetime by 3.5 hours and 2 minutes for fine and coarse particles respectively in comparison to DUST4. Consequently, the dust column burden increases by 4% (fine) and 3% (coarse) that boosts DOD by approximately 10% over the desert and the Mediterranean.

The negative surface radiative forcing of DUST12 experiment in the shortwave spectrum is regionally enhanced by -
0.5W·m-2 while the top of the atmosphere radiative forcing is intensified by almost -1W·m-2 (10%). The negative changes in the surface radiative forcing are concentrated in the western part of Africa and extend over the Eastern Atlantic, where the near-constant annual dust plume is located. On the longwave spectrum DUST12 experiment enhances the positive RF at the surface and TOA by 0.1W·m-2 in a large portion of the Sahara desert, the northern part of the Arabian Peninsula and the middle east. Although the radiation transfer scheme RRTM, known for its detailed integration on the longwave spectrum,
revealed that these longwave RF changes can be also locally negative and therefore smaller when spatially averaged over the Sahara desert.

Overall, this study highlights that the radiative differences between the two dust size bin treatments are relatively small. The simulated shortwave radiative forcing by the 4-bin isolog method is to some extent numerically efficient and acceptable. Nevertheless, our work emphasize that the simplified representation of the 4-bin approach underestimates the direct radiative
forcing, a fact that should be taking into account by future researches that study the same region. We have to note that other typical size distribution (e.g. Alfaro et al. 1998; Zender et al. 2003) could be much more sensitivity to the binning partitioning method and number which will generate bigger changes in dust column burden DOD and RF.

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





**Figure 1: Dust bin specific extinction coefficient, single scattering albedo (SSA) and particle asymmetry parameter on the 350-640nm spectral band. Red color depicts the 4 and blue the 12 bin experiment. Top and bottom axis depicts the dust particle diameter limits of the 4 and 12 dust size bins respectively.**



**Figure 2: RegCM4 simulated domain (black box) and the subregions selected for the analysis (blueboxes). Background colors depict the desert (dark brown) and semi-desert (light brown) grid cells assign by the model. Dashed lines illustrates the simulated topography used on RegCM4 in meters.**





**Figure 3: Dust optical depth of the DUST4 experiment and the LIVAS dust product for the period January 2007 to November 2014.**

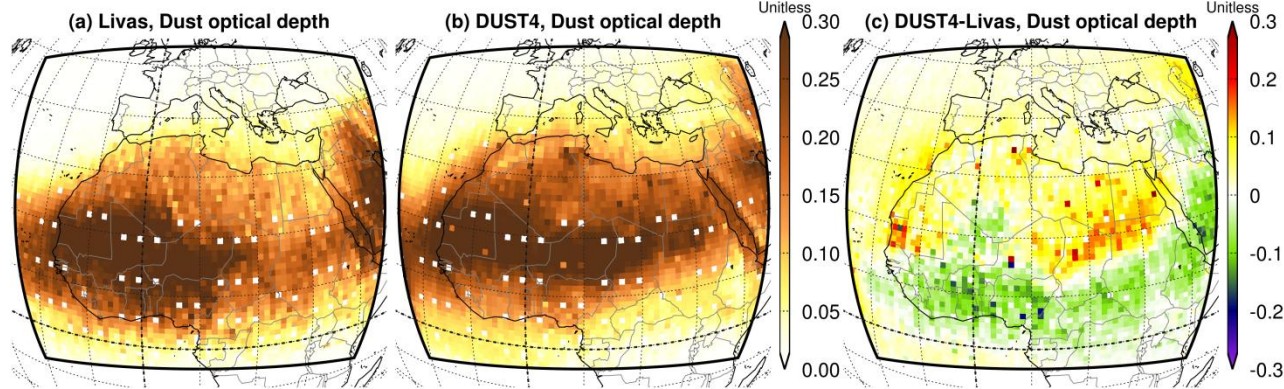

5    **Figure 4: Dust optical depth annual cycle of LIVAS and DUST4 experiment for the period January 2007 to November 2014. Shaded areas display the 95% confidence interval of the mean.**



**Figure 5:** Coarse dust particle column burden of the DUST4 experiment (lines) alongside with the production/destruction column tendencies (bars) averaged for the period December 2006 to November 2014. Shaded areas display the 95% confidence interval of the mean.




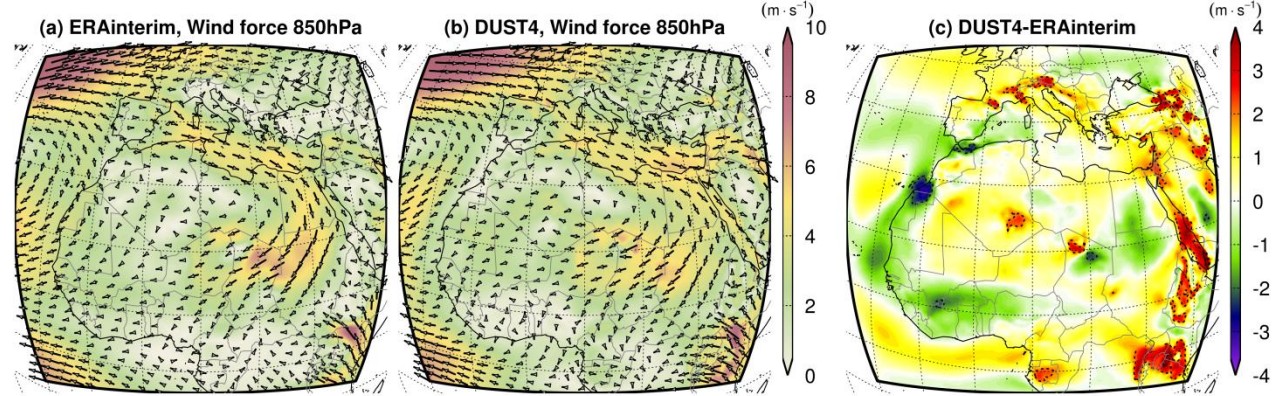

**Figure 6: Fine dust particles column burden of the DUST4 experiment (lines) alongside with production/destruction column tendencies (bars) averaged for the period December 2006 to November 2014. Shaded areas display the 95% confidence interval of the mean.**

**Figure 7: Wind velocity at 850hPa of the DUST4 experiment against the reanalysis ERA-interim for the period December 2006 to November 2014.**




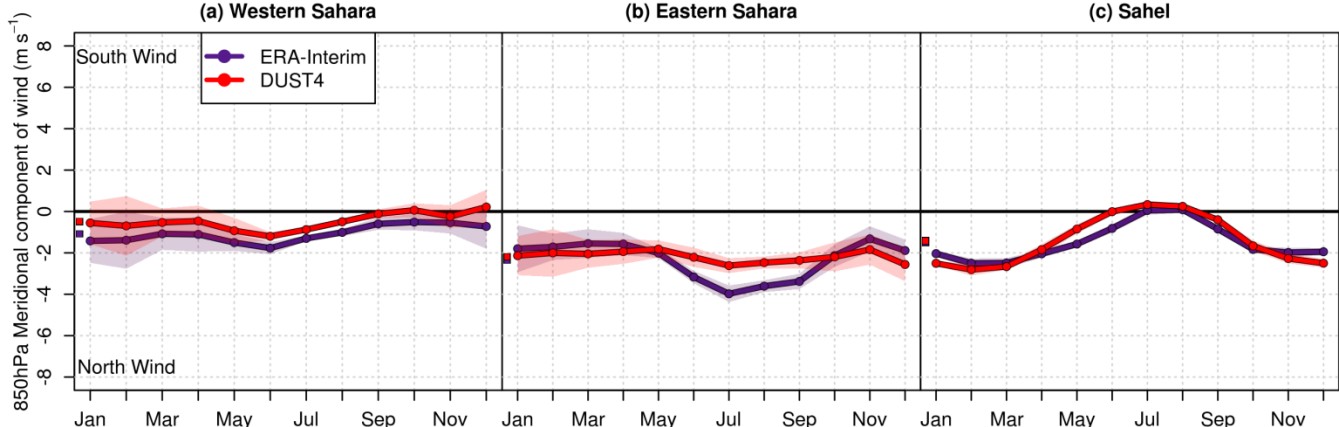

**Figure 8:** Annual cycle of the meridional wind component of ERA-interim and DUST4 experiment for the period December 2006 to November 2014. Positive and negative values indicate south and north wind respectively. Shaded areas display the 95% confidence interval of the mean.

**Figure 9:** Total precipitation annual cycle of the ERA-interim and DUST4 experiment for the period December 2006 to November 2014.





**Figure 10: Dust extinction profiles of LIVAS and DUST4 experiment. The percent bias (P.Bias) of dust extinction located between 0.2-5km, 5-10km and 10-20km for each subregion is illustrated on the plots. The bars depict the percent of the vertical tendencies for 0.2-5km and 5-10km.**



Figure 11: Dust optical depth, coarse (>2.5µm) and fine (<2.5µm) column burden of DUST4 and DUST12 experiments for the period December 2006 to November 2014.





**Figure 12:** Annual cycle of dust optical depth of DUST12 and DUST4 experiment for the period December 2006 to November 2014.



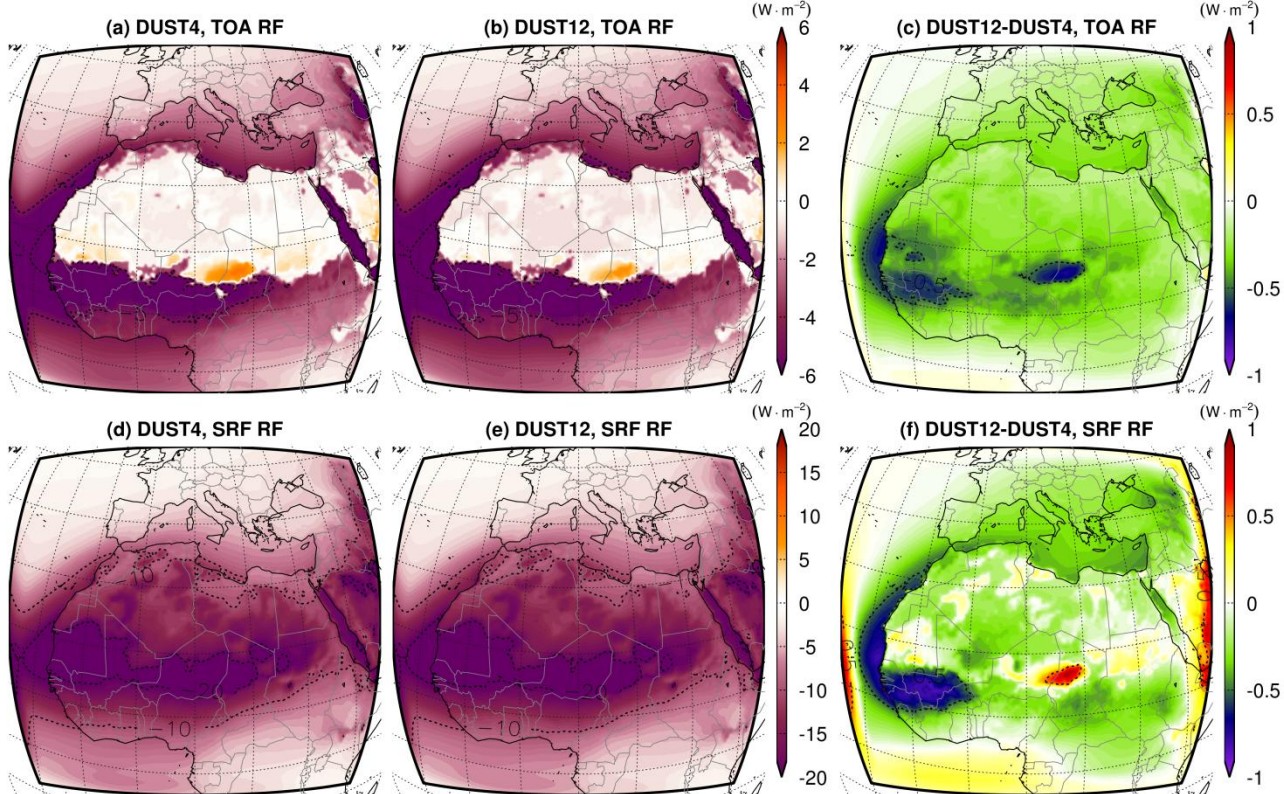

Figure 13: Top of the atmosphere (top) and surface (bottom) radiative forcing on the shortwave spectrum for the period December 2006 to November 2014.





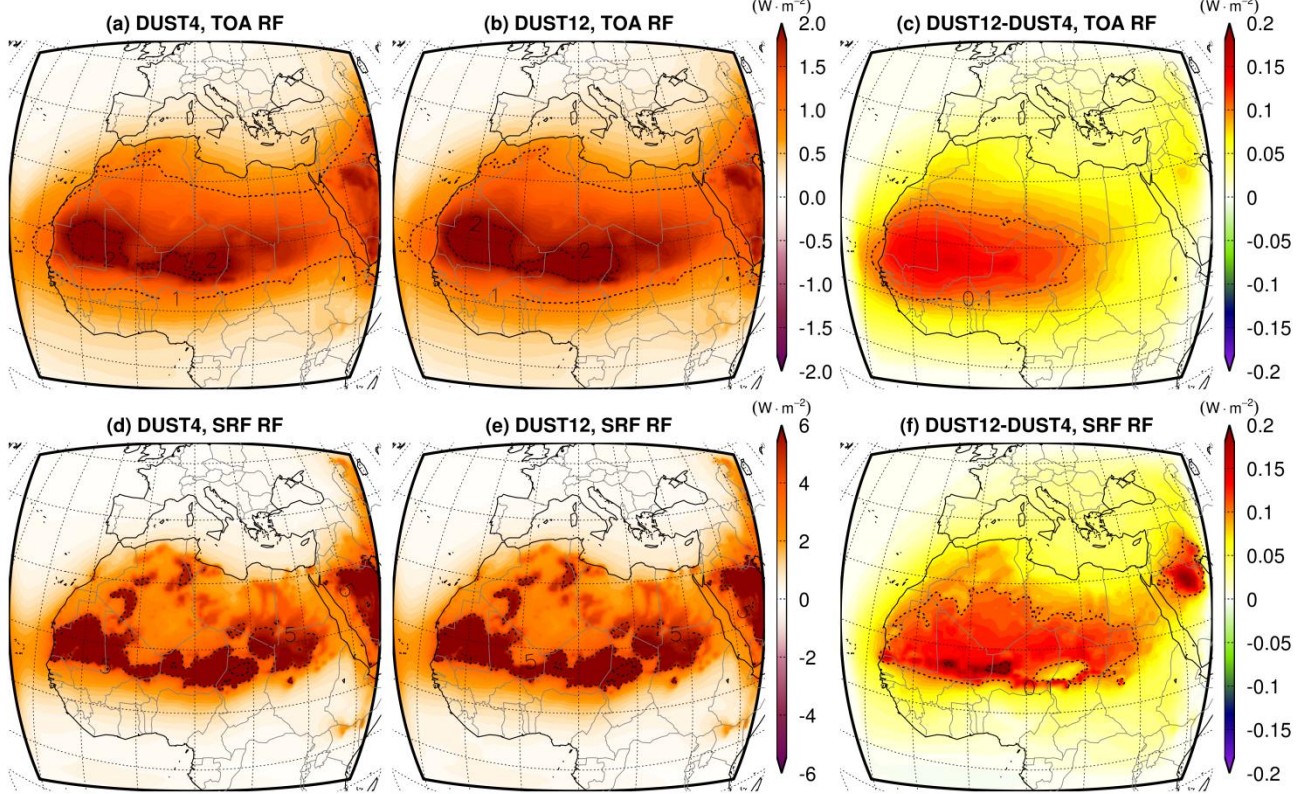

Figure 14: Top of the atmosphere (top) and surface (bottom) radiative forcing on the longwave spectrum for the period December 2006 to November 2014.