# Peer review of "Impact of dust size parameterizations on aerosol burden and radiative forcing in RegCM4"

_Atmospheric Chemistry and Physics, 2016_

## Referee Comment (RC1) · Anonymous Referee #2 · 23 Aug 2016

Review on the "Dust size parameterization in RegCM4: Impact on aerosol burden and radiative forcing" by Tsikerdekis et al.

Size distribution play a role in dust aerosol emission, deposition, optical properties, and radiation. This paper examines the sensitivity of the 12-bin and 4-bin approaches in the regional climate model (RegCM4). The experiment has been applied over the North Africa which is the most important source area for the period 2007-2014. The model results are compared with the LIdar climatology of Vertical Aerosol Structure for Space-based lidar simulation studies (LIVAS). LIVAS AOD is derived from CALIPSO AOD observations and its uncertainty is +/-0.1 compared with AEROENT. In this study RegCM4 found that 12-bin approach results small increase in dust loading (+3~+4%) and larger increase in DOD (10%). Its Shortwave Radiative Forcing is -8~-10% (-0.18~-0.24 W/m2) stronger than 4-bin approach.

I also agree with the paper on the technical side, i.e., the more size bins can provides better information thus it may improve the model results. The presentation is well organized and good. However, I have a major concern that the model sensitivity results in this paper is more about technical aspect of modeling process, not presenting new concept or method. That is why the evaluation in section 3.1 (including eight figures) is conducted solely with the 4-bin method. Thus I have to conclude that this paper does not make a substantial contribution to scientific progress within the scope of Atmospheric Chemistry and Physics. I suspect this paper may be more suitable for other technical modeling journals.

Comments:

1. The paper does not present much about emission which is heavily affected by size distribution. Can you show more about dust emission, e.g., spatial distributions and size distributions?

2. P1, L20-21 and P3, L18-19: Please consider removing "minimize the error". More size bins shall provide more information but it does not guarantee if it is more realistic than less bin numbers. Both 12-bin and 4-bin approaches are globally homogeneous and both have large uncertainties. This study shows that 12-bin results some changes in dust loading (3-4%) and DOD (10%). Since the model sensitivity is small or moderate it needs statistical significance test to prove the improvement in 12-bin approach. Emission, DOD, and/or RF from the new model need to be compared with observations to show how much bias is reduced compared to the old model.

3. P5, L20: Please clarify what is a new dust scheme. Size distribution only or new scheme?

4. P6, L31: Please provide the old and new lidar ratio values.

5. P7, L8-11: The sentence is unclear, since the study domain is over land. I believe MODIS Deep Blue AOD is well validated with AERONET data over the source region (please check with the NASA Deep Blue website and other documentations).

6. P8, L15: Please be specific the size is radius or diameter.

7. P9, L5 and Figure 2: About one half of the Sahel is desert and the other half is non-source according to the map. Semi-arid source is only small fraction. Bodele is in Sahel. It is unclear what are the characteristics of the Sahel in this study. The seasonality of wind, precipitation, loading, and DOD over the Sahel is mixed with Sahara and Savanna.

8. P13, L34-P14, L5: The sentences do not belong to the results. Please consider to relocate or remove.

9. P16, L1-2: Unclear sentence. Please consider to remove it.

10. P16, L14-30: Again, it does not belong to result section.

---

## Referee Comment (RC2) · Anonymous Referee #3 · 4 Nov 2016

**Review on the "Dust size parameterization in RegCM4: Impact on aerosol burden and radiative forcing" by Tsikerdekis et al.**

Authors of the manuscript present an interesting analyzis of which processes control the dust aerosol load over the Sahara and surrounding region by 1) performing a detailed validation againts available aerosol observational products and 2) by performing sensitivity test of the model performance on the choice of the aerosol size distribution discretization method. Although the amount of work done is rather small (validation of the default model configuration and a comparison to a more complex parameterization), the results are discussed in very detailed way with lots of aspects

considered. In this view, it represents a valuable contribution to the aerosol science and to modeling communities focusing to aerosol regional transport and its model representation. Thus I recommend it to publication in ACP after considering the following comments:

General comments:

In principle, the authors present and compare three datasets: observation and two model realizations. I suggest to join all three datasets in the validation part. It makes the reader easier to see the difference between the two aerosol size bin discretization schemes in the view of the reference observational data. Of course, this holds only for those quantities, where observations are available.

The presented changes caused by the introduction of new aerosol size bin scheme are rather small. I agree with the other reviewer, that this requires a statistical significance test of the differences.

Specific comments and corrections:

Pg 1, line 14 (Abstract): It is not the sensitivity of the parameterization, but the sensitivity of the model representation of aerosol on the choice of the parameterization. Rephrase, please.
Pg 1, line 20: Change 'minimize' to 'reduces'.
Pg 1, line 31: change 'taking' to 'taken'.
Pg 3, lines 15-16: This statement is true, however please add 1-2 sentences on how this number affects the modeling of aerosols, at least in theory.
Pg 4, line 19: change 'to improve radiation processes' to 'to improve the model representation of the radiation processes...'
Pg 4, line 23: use rather 'driven by RegCM4 meteorology, while ...'

Pg 5, line 8: remove 'is' ('used' only, not 'is used')

Pg 5, lines 25-26: So aerosols cannot move from one bin to another meaning that there is no aerosol fragmentation or aerosol coagulation?

Pg 6, lines 5-7: I would be interesting to compare the presented optical properties with those obtained by using only one effective diameter – i.e. the one that describes the bin (the center of it).

Pg 9, line 7: correct 'buns' to 'bins'

Pg 16,line 8: correct 'Thus is exhibits' to 'Thus it exhibits..'

Pg 17, line 2: ..on emissions? I missed something in the manuscript? At this point, I would recommend to present an emission figure to gain some idea about its spatial distribution.

Pg 18, line 14: The use of word 'underestimates' is not correct, as we have no comparison of RF figures with observation: there is no information on the reference values of the RF, so it can possibly be that the 4 bin approach is closer to the reality.

---

## Author Response (AR3)

**1st Review**

Response to **Anonymous Referee #2** concerning the paper *"Dust size parameterization in RegCM4: Impact on aerosol burden and radiative forcing"* (http://www.atmos-chem-phys-discuss.net/acp-2016-434).

November 30, 2016
Dear editorial and respected reviewer,

Thank you very much for reviewing our manuscript and providing us with such a constructive feedback. We believe that your comments helped us highlight some critical aspects of the paper and add important content which elevates the quality of our work. All the short comments and typographical errors were corrected in the new version of the paper, while comments that needed more explanation and additional material are discussed below. The structure of the responses includes (1) comments from Referee, (2) a detailed response (3) changes implemented in the new version of the paper. Quotes from the initial version of the paper are highlighted with pale red along with the corresponding page and lines. Quotes from the new version of the paper are highlighted with pale blue along with the corresponding page and lines. The specific phrases changed-modified are underlined for convenience. In cases where the quotes are discussed but not changed are highlighted with grey. I sincerely hope you will be fully satisfied with all the changes we have made.

Best Regards,
Athanasios Tsikerdekis
* * *
1. The paper does not present much about emission which is heavily affected by size distribution. Can you show more about dust emission, e.g., spatial distributions and size distributions?

This is an important issue that you have raised and we sought to address in the paper. We agree it adds value and it will additionally help the readers understand the work better. The Fine (DUST01+DUST02) and Coarse (DUST03+DUST04) emission tendencies (kg kg$^{-1}$ s$^{-1}$) are shown in Figure 5 and Figure 6 as monthly values averaged over a subregion for the DUST4 experiment. But of course the spatial distribution of emission flux (mg m$^{-2}$ day$^{-1}$) might be helpful for the reader so we are adding the following figures (emission, dry deposition, wet deposition) as supplementary material. Note that the spatial distribution of emission and deposition fluxes are almost identical between the fine and the coarse dust particles, although the coarse particles deposition and emission fluxes are ten times greater. Also, the following comment was added in the main paper.

P12, L4-5: The spatial distribution of wet deposition, dry deposition and surface emission fluxes is depicted for the fine and coarse particles in FigureS 3.

[Figure]

**FigureS 3: Wet deposition, dry deposition and surface emission fluxes of fine (a, b, c) and coarse (d, e, f) dust particles in DUST4 experiment for the period December 2006 to November 2014.**
* * *
2. P1, L20-21 and P3, L18-19: Please consider removing "minimize the error". More size bins shall provide more information but it does not guarantee if it is more realistic than less bin numbers. Both 12-bin and 4-bin approaches are globally homogeneous and both have large uncertainties. This study shows that 12-bin results some changes in dust loading (3-4%) and DOD (10%). Since the model sensitivity is small or moderate it needs statistical significance test to prove the improvement in 12-bin approach. Emission, DOD, and/or RF from the new model need to be compared with observations to show how much bias is reduced compared to the old model.

As you correctly point out, the phrase "minimize the error" is too generic and can be misinterpreted as "minimizing the bias in comparison to observations", which is not always the case. Therefore, we have removed and rephrased it respectively.

P1, L19-21: Increasing the number of transported dust size bins theoretically improves the representation of the physical properties of dust particles within the same size bin. Thus, more size bins minimize the error and improve the simulation of atmospheric processes.

P1, L20-22: Increasing the number of transported dust size bins theoretically improves the representation of the physical properties of dust particles within the same size bin. Thus, more size bins  improve the simulation of atmospheric processes.

P3, L18-19: The greater number of dust size bins minimizes model error especially for particle dry deposition and thus allows to more accurately simulate both the atmospheric dust burden and the interaction with radiation (Foret et al., 2006; Menut et al., 2007).

P3, L22-23: Greater number of dust size bins  improves particle dry deposition and thus allows to more accurately simulate both the atmospheric dust burden

and the interaction with radiation (Foret et al., 2006; Menut et al., 2007).

More size bins provide more information than less bin numbers. Furthermore, the isolog partitioning method is arbitrary and mathematical, with no connection to the physical world processes. Isogradient partitioning method of the bins takes into account the deposition velocity of dust particles according to their size. Considering both the number and the partitioning method we are highlighting that the 12bin-isogradient is actually more "realistic" than the 4bin-isolog method (closer to reality in terms of the physical processes that take place in the atmosphere). Of course this improvement does not guarantee that is going to fix the biases that originate from other processes as already stated in the paper:

Pg 14, lines 30 - Pg 15, lines 5: Previous studies (Foret et al., 2006; Menut et al., 2007) revealed that the new approach in dust size bin partitioning and number (which is also adopted in our DUST12 experiment) more realistically simulates the transport and dry deposition of the dust size bins. This does not imply that the simulated biases will be reduced for all regions in the DUST12 experiment. The major factor that controls and potentially creates the first order biases in dust models is the balance between the emission and sedimentations terms (e.g. Figure 5, Figure 6). As discussed in Section 3.1, some positive DOD biases (e.g. Eastern Sahara) might be due to an underestimation of the outflow of dust or a local overestimation of the emission flux from surface wind velocity errors. Therefore, although the new dust size parameterization theoretically improves dry deposition, it does not necessarily regulate or improve the biases that originate from other processes.

Thank you for highlighting the statistical significance issue. We have calculated the statistical significance of the differences using the monthly data for each grid point for the DOD, dust column burden and radiative forcing using the two-tailed paired t-test. In all variables and almost in all grid points the differences are statistically significant at the 95% confidence level (p.value < 0.05). We have also updated the text in the new version of the paper to highlight this issue. The plots shown below were not updated in the paper since we believe that they do not offer additional information to the reader; the vast majority of the grid points show differences that are statistically significant (shaded areas in the differences), hence we are just mentioning this in the text.

Pg 14, lines 13-20: The DOD percent increase is between 10.4% and 13% for all the subregions. Furthermore, there is a distinctive increase by 0.04 with the 12-bin model over the Sahara desert and especially along the Sahel region where the DOD values are higher (Figure 11c). In comparison with the DUST4 simulation the DUST12 simulation increases the deposition lifetime (column burden/total deposition flux) by 3.5 hours and 2 minutes for fine and coarse particles respectively. Consequently, that increases the dust column burden of fine (+4%) and coarse (+3%) particles (Figure 11f,i). The changes in the fine particles correlates better with the changes in DOD, because dust extinction coefficient is much higher for fine particles (<2.5µm) (Figure 1). Over the Middle East and the northern part of the Arabian Peninsula we observe a distinct increase on the coarse dust column burden by 10mg•m-2.

Pg 15, lines 17-26: The DOD percent increase is between 10.4% and 13% for all the subregions. Furthermore, there is a distinctive increase by 0.04 with the 12-bin model over the Sahara desert and especially along the Sahel region where the DOD values are higher (Figure 11c). In comparison with the DUST4 simulation the DUST12 simulation increases the deposition lifetime (column burden/total deposition flux) by 3.5 hours and 2 minutes for fine and coarse particles respectively. Consequently, that increases the dust column burden of fine (+4%) and coarse (+3%) particles (Figure 11f,i). The changes in the fine particles correlates better with the changes in DOD, because dust extinction coefficient is much higher for fine particles (<2.5µm) (Figure 1). Over the Middle East and the northern part of the Arabian Peninsula we observe a distinct increase on the coarse dust column burden by

10mg•m-2. The differences of DOD and column burden between the two experiments, calculated from the monthly data for each grid, are statistically significant at the 95% confidence level according to a two-tailed paired t.test for almost all the grid points of the simulated domain.

Pg 16, lines 11-13: Similarly, at the surface the positive radiative forcing rise by 0.08W·m-2 (3.0%) and 0.09W·m-2 (2.7%) in eastern and western Sahara and 0.9W·m-2 (6.3%), 0.08W·m-2 (4.8%) and 0.9W·m-2 (6.5%) in western, central and eastern Mediterranean (Figure 14f).

Pg 17, lines 16-20: Similarly, at the surface the positive radiative forcing rise by 0.08W·m-2 (3.0%) and 0.09W·m-2 (2.7%) in eastern and western Sahara and 0.9W·m-2 (6.3%), 0.08W·m-2 (4.8%) and 0.9W·m-2 (6.5%) in western, central and eastern Mediterranean (Figure 14f). The shortwave and longwave radiative forcing differences between the two experiment, calculated from the monthly data for each grid, are statistically significant at the 95% confidence level according to a two-tailed paired t.test for almost all the grid points of the simulated domain.

[Figure]

[Figure]

Thank you for highlighting the emission, DOD and RF evaluation again. Let us explain the case in each one of them below. Dust emission is identical between the two experiments (DUST4, DUST12) and this was done on purpose (as already stated in Section 2.5 Experimental set-up) in order to compare the

size bin changes in the two experiments. The two experiments have the same meteorological fields, the same wind fields, thus the same emission fluxes. A direct evaluation of the emission fluxes with observations is not available at this spatial scale, thus we evaluate and discuss emission flux indirectly by using the surface wind speed from the ERA-interim in the Section "3.1 Evaluation".

Dust Optical Depth (DOD) is already being evaluated for both experiments with the state-of-the-art as of right now dust product LIVAS (FigureS 5). FigureS 5 was added due to the comments that we have received in the 1st review phase. As already discussed in the paper (Section "3.2 Comparison of 4-bin and 12-bin experiments"), the reduction of bias in comparison to observation doesn't necessarily mean that we are improving the physics in the model. DUST4 experiment exhibits lower bias in comparison to DUST12 experiment in most regions since conflicting processes that are overestimated-underestimated in the model (e.g. wind fields, precipitation, surface characteristics) might give a DOD closer to the observational data, but for the wrong reasons.

To our knowledge there are no observational data for Dust Radiative Forcing (DRF). But evaluating DOD is the next best thing, since DOD is the extinction of radiation by the dust particles in the atmosphere. Of course since DRF is also a function of the simulated SW and LW radiation, we have added as supplementary material the following plots which evaluate the net ShortWave (SW) and LongWave (LW) in the SuRFce (SRF) using the CERES measurements. We also added the following phrase at the end of the Section "3.1 Evaluation".

P15, L7-9: The model was evaluated also in terms of the surface net downward shortwave and net upward longwave radiation flux against CERES satellite measurements (FigureS 8). In the shortwave spectrum radiation flux bias ranges between -10Wm$^{-2}$ and 10Wm$^{-2}$ above the desert and the Mediterranean, while in the longwave the differences are mostly positive (~10Wm$^{-2}$).

[Figure]

**FigureS 8: Net shortwave downward (a, b, c) and net longwave upward flux (d, e, f) in the surface**

**of CERES and DUST4 experiment for the period December 2006 to November 2014.**
* * *
3. P5, L20: Please clarify what is a new dust scheme. Size distribution only or new scheme?

Thank you for pointing this out. We have rephrased the following sentence.
P5, L19-20: Following the methodology of (Foret et al., 2006), we have implemented a new dust scheme that resolves twelve size bins instead of four.
P5, L25-27: Following the methodology of (Foret et al., 2006), we have implemented a new dust size discretization scheme that resolves twelve dust transport size bins instead of four.
* * *
4. P6, L31: Please provide the old and new lidar ratio values.

Thank you for highlighting this issue, I think is of great importance therefore we have added the Lidar Ratio (LR) values that was used prior in the CALIPSO measurements and the regional specific new LR values used in the LIVAS product in the last sentence. Plus in the supplementary we have added the following plot to show the spatial distribution of LR assumption.
P6, L31 - P7, L4: The LIVAS extinction dust product is corrected for the Lidar Ratio (LR) based on multi-year measurements performed by the ground-based lidar stations of the EARLINET lidar network (https://www.earlinet.org). The LR of dust particles depends on their refractive index and may vary for aerosols of the same type. The refractive index values rely upon the composition of dust and most importantly on the relative proportion of clay-sized mineral illite in dust (Schuster et al., 2012). Thus, regions with different physiochemical dust characteristics leads to different LR values. The 0.3.1 version of LIVAS separates the globe into three regions, specified based on known dust sources and loadings with specific physio-chemical composition and LR for each region.
P7, L4-L12: The LIVAS extinction dust product is corrected for the Lidar Ratio (LR) based on multi-year measurements performed by the ground-based lidar stations of the EARLINET lidar network (https://www.earlinet.org) and intensive campaigns in different dust regions arround the globe (Wandinger et al., 2010; Hänel et al., 2012; Baars et al., 2016). The LR of dust particles depends on their refractive index and may vary for aerosols of the same type. The refractive index values rely upon the composition of dust and most importantly on the relative proportion of clay-sized mineral illite in dust (Schuster et al., 2012). Thus, regions with different physiochemical dust characteristics leads to different LR values. The 0.3.1 version of LIVAS separates the globe into three regions, specified based on known dust sources and loadings with specific physio-chemical composition and LR for each region. The globally LR value 40sr used in CALIPSO retrieval algorithm (Omar et al., 2009) was replaced with region specific LRs of 40sr, 50sr and 55sr (FigureS 2).

[Figure]

**FigureS 2: The region specific Lidar Ratio assumption used in LIVAS.**
* * *
5. P7, L8-11: The sentence is unclear, since the study domain is over land. I believe MODIS Deep Blue AOD is well validated with AERONET data over the source region (please check with the NASA Deep Blue website and other documentations).

We agree with the reviewer and indeed we believe the new changes help situate the case better. The retrieval accuracy and the spatial coverage of the MODIS Deep Blue AOD has been improved in the latest version C6 in comparison to the C5 version. Future studies will determine the correlation between MODIS Deep Blue (C6) AOD and the LIVAS DOD. Therefore, we are highlighting in the text the version of MODIS (C5) evaluated with LIVAS and the need for future comparison with the new MODIS Deep Blue (C6) AOD. We feel the need again to note that the LIVAS post-processing analysis provides the DOD (and not AOD like MODIS) which can be directly used to evaluate the DOD of the model in this paper.

P7, L5-13: LIVAS has been evaluated against AERONET stations globally by Amiridis et al. (2015). The results show that the aerosol optical depth differences are between ±0.1 in most cases. Over the southwestern Sahara desert, LIVAS underestimates the AERONET AOD by -0.1, and this bias may be related with the dust underestimation of CALIPSO found in previous studies (Amiridis et al., 2013; Schuster et al., 2012; Tesche et al., 2013; Wandinger et al., 2010). Amiridis et al. (2013) showed that LIVAS correlates well with the Dark Target MODIS retrieval over sea, yet the correlation between MODIS Deep Blue and LIVAS over Sahara are weak (results not shown). This could be attributed to the Deep Blue MODIS retrieval, which uses passive remote sensors for dust aerosol optical depth that take into account numerous assumptions (e.g. high reflectivity). Thus, LIVAS is a more reliable product over the deserts with higher accuracy than the products coming from passive remote sensing techniques.

P7, L13-L21: LIVAS has been evaluated against AERONET stations globally by Amiridis et al. (2015). The results show that the aerosol optical depth differences are between ±0.1 in most cases. Over the southwestern Sahara desert, LIVAS underestimates the AERONET AOD by -0.1, and this bias may be related with the dust underestimation of CALIPSO found in previous studies (Amiridis et al., 2013; Schuster et al., 2012; Tesche et al., 2013; Wandinger et al., 2010). Amiridis et al. (2013) showed that

LIVAS correlates well with the Dark Target MODIS retrieval over sea, yet the correlation between MODIS Deep Blue (version C5) and LIVAS over Sahara are weak (results not shown). The new version of MODIS Deep Blue (C6) improved its retrieval accuracy and spatial coverage globally (Sayer et al., 2014, 2015) and the Mediterranean region (Georgoulias et al., 2016a) in comparison to AERONET. Thus, further research is needed to determine the correlation between LIVAS DOD and MODIS Deep Blue (C6) AOD.
* * *
6. P8, L15: Please be specific the size is radius or diameter.

Thank you for pointing out this clarification. It is diameter.

P8, L15-16: Using a simple one-dimensional box model they simulated an experiment with a detailed particles size distribution that used 1000 size bins within the range of 0.001-100μm.

P8, L24-25: Using a simple one-dimensional box model they simulated an experiment with a detailed particles size distribution that used 1000 size bins within the range of 0.001-100μm diameter.
* * *
7. P9, L5 and Figure 2: About one half of the Sahel is desert and the other half is non source according to the map. Semi-arid source is only small fraction. Bodele is in Sahel. It is unclear what are the characteristics of the Sahel in this study. The seasonality of wind, precipitation, loading, and DOD over the Sahel is mixed with Sahara and Savanna.

We agree with the reviewer. The rough delimitation of the box-shaped subregions may affect the levels (but as it was found not the seasonality) of the dust column burden, DOD, wind velocity and precipitation annual cycles (for example DOD of Sahel). Therefore, we have re-organized all the subregions making them more representative for dust fluxes and concentration as well as for the meteorological variables.

The new delimitation of each region is shown in the following plot. Sahel was delimited using the CRU precipitation for the period 2001-2014. The selected grid cells for the Sahel receive annual precipitation between 100mm and 600mm (Ali and Lebel, 2009; Nicholson, 2013) and now Bodele Depression belongs to the Eastern Sahara subregion. Eastern and Western Sahara were selected according to the desert and semi-desert landuse assigned by the model. Their southern border was masked by the Sahel grid points. The three Mediterranean subregions now contain only non-desert grid points, while their western, northern and eastern boundaries were kept the same. Due to these changes some plots were updated accordingly in the final manuscript: Figure 4, Figure 5, Figure 6, Figure 8, Figure 9, Figure 10, Figure 12 as well as Figure S4 and Figure S5 (which are FigureS7 and FigureS9 in the new version of the paper). The following paragraph was added in the Section "2.5 Experimental set-up" to explain the delimitation of each region.

P9, L15-22: The simulated domain was separated into six distinct subregions: Sahel, Eastern Sahara (ESah), Western Sahara (WSah), Eastern Mediterranean (EMed), Central Mediterranean (CMed) and Western Mediterranean (WMed) (Figure 2b). Sahel was delimited using CRU precipitation for the period 2001-2014. The selected grid cells receive annual precipitation between 100mm and 600mm (Ali and Lebel, 2009; Nicholson, 2013) and are located in the southern border of Sahara. Eastern and

Western Sahara were selected according to the desert and semi-desert landuse assigned by the model. Their southern borders were masked by the Sahel grid points. The three Mediterranean subregions contain only non-desert grid points and are separated according to their DOD seasonality (Israelevich et al., 2012).

[Figure]

**Figure 2: (a) The desert (dark brown) and semi desert (light brown) grid cells assign by the model along with the simulated topography used on RegCM4 in meters. The black box depicts the simulated domain. (b) The distinct six subregions used in the analysis (details in text).**
* * *
8. P13, L34-P14, L5: The sentences do not belong to the results. Please consider to relocate or remove.

Thank you for raising this issue and we can understand your concern. The structure of our paper is consisted by a combined Results+Discussion Sections (3.1, 3.2, 3.3). Thus, we have changed the title of the Section "3 Results" into "3 Results and Discussion". We believe that this structure greatly helps the flow of the paper without making it too long. Thus, we believe that the following lines belong to the Results and Discussion section, since they are concluded from the analysis of Figure 10 (Dust Extinction evaluation). Prior to this section we are explaining what processes govern dust vertical distribution in each altitude and in (P13, L34-P14, L5) we are suggesting possible reason for an overestimation in the mid Troposphere by the model.

P13, L34-P14, L5: Considering the overestimated DEX profile in this altitude range it suggests that the cumulus convection activity/convective transport mechanism is overactive in Tiedtke scheme (Tiedtke, 1989) or another negative sign process (e.g. sedimentation) is not properly represented by the model. Discrepancies in the vertical distribution of dust can also be misinterpreted from possible local emission errors, the lack of simulated vertical levels or wet deposition biases. Further research is needed in this regard to reduce the mean vertical distribution bias of dust in the model.
* * *
9. P16, L1-2: Unclear sentence. Please consider to remove it.

Indeed, the following sentence was too generic, therefore we have removed it. Thank you for indicating

this.
* * *
10. P16, L14-30: Again, it does not belong to result section.

Thank you for brining this up, though let me elaborate on it. As already noted in comment 8, we have written our Section "3 Results and Discussion", since it helps the reading flow. Thus, the following paragraph belongs to the resul-discussiont section since we are presenting and discussing results between four 1-yearly experiments in order to more accurately validate our radiative forcing results in the longwave spectrum. All the plots presented (FigureS 8 and FigureS 9) are analyzing the radiative forcing of dust in the longwave spectrum, thus we have put them in the section "3.3 Radiative Forcing".

P16, L14-30: However, the current treatment of the optical properties of dust in the longwave spectrum of CCM3 scheme is limited and does not account for specific absorption coefficient for each dust size bin. Furthermore, the CCM3 longwave bands are concentrated on the absorption of $H_2O$ and $CO_2$ and they do not integrate in detail the absorbing aerosol component part of the longwave. Therefore, we have conducted two similar dust experiments for June 2008 using the radiation transfer scheme RRTM, which is known for its detailed longwave calculation. We note again that the current version of RRTM (+McICA) produces a random generated noise on radiation fields, therefore the dust emission fluxes are not identical between the DUST4 and DUST12 experiments. Our preliminary simulations using the RRTM scheme (FigureS 8), have shown notably lower longwave dust RF over the desert in comparison to CCM3 (FigureS 7) for the same period and roughly the same DOD levels and spatial patterns. More specifically, the DOD and the longwave radiative forcing averaged above the Sahara for the CCM3_DUST4 experiment is DOD=0.31, SRF=3.5W·m-2 and TOA=2.1W·m-2. While the related values for the RRTM_DUST4 experiment is DOD=0.29, SRF=1.9W·m-2 and TOA=0.6W·m-2. Although DOD increases in the 12 bin experiment in both radiation schemes (FigureS 7c, FigureS 8c), the longwave RF changes display a striking difference. With the CCM3 radiation scheme, the longwave radiation linearly increases with higher DOD and dust burden values (FigureS 7f and i). In comparison, the RRTM scheme uses specific absorption coefficient for each dust size bin, taking into account the fine/coarse dust burden changes. Thus is exhibits local increases or decreases of the longwave RF (FigureS 8f and i), according to the changes of fine to coarse dust burden. Overall the total longwave RF increase in RRTM is smaller compared to CCM3 when spatially averaged over the Sahara desert.
* * *
References Added.

Ali, A. and Lebel, T.: The Sahelian standardized rainfall index revisited, Int. J. Climatol., 29(12), 1705–1714, doi:10.1002/joc.1832, 2009.

Baars, H., Kanitz, T., Engelmann, R., Althausen, D., Heese, B., Komppula, M., Preißler, J., Tesche, M., Ansmann, A., Wandinger, U., Lim, J.-H., Ahn, J. Y., Stachlewska, I. S., Amiridis, V., Marinou, E., Seifert, P., Hofer, J., Skupin, A., Schneider, F., Bohlmann, S., Foth, A., Bley, S., Pfüller, A., Giannakaki, E., Lihavainen, H., Viisanen, Y., Hooda, R. K., Pereira, S. N., Bortoli, D., Wagner, F., Mattis, I., Janicka, L., Markowicz, K. M., Achtert, P., Artaxo, P., Pauliquevis, T., Souza, R. A. F., Sharma, V. P., van Zyl, P. G., Beukes, J. P., Sun, J., Rohwer, E. G., Deng, R., Mamouri, R.-E. and Zamorano, F.: An overview of the first decade of PollyNET: an emerging network of automated Raman-polarization lidars for continuous aerosol profiling, Atmos. Chem. Phys., 16(8), 5111–5137, doi:10.5194/acp-16-5111-2016, 2016.

Georgoulias, A. K., Alexandri, G., Kourtidis, K. a., Lelieveld, J., Zanis, P. and Amiridis, V.: Differences between the MODIS Collection 6 and 5.1 aerosol datasets over the greater Mediterranean region, Atmos. Environ., 147, 310–319, doi:10.1016/j.atmosenv.2016.10.014, 2016.

Hänel, A., Baars, H., Althausen, D., Ansmann, A., Engelmann, R. and Sun, J. Y.: One-year aerosol profiling with EUCAARI Raman lidar at Shangdianzi GAW station: Beijing plume and seasonal variations, J. Geophys. Res. Atmos., 117(D13), n/a–n/a, doi:10.1029/2012JD017577, 2012.

Nicholson, S. E.: The West African Sahel: A Review of Recent Studies on the Rainfall Regime and Its Interannual Variability, ISRN Meteorol., 2013, 1–32, doi:10.1155/2013/453521, 2013.

Sayer, A. M., Hsu, N. C., Bettenhausen, C., Jeong, M.-J. and Meister, G.: Effect of MODIS Terra radiometric calibration improvements on Collection 6 Deep Blue aerosol products: Validation and Terra/Aqua consistency, J. Geophys. Res. Atmos., 120(23), 12,157–12,174, doi:10.1002/2015JD023878, 2015.

Sayer, A. M., Munchak, L. A., Hsu, N. C., Levy, R. C., Bettenhausen, C. and Jeong, M.-J.: MODIS Collection 6 aerosol products: Comparison between Aqua's e-Deep Blue, Dark Target, and "merged" data sets, and usage recommendations, J. Geophys. Res. Atmos., 119(24), 13,965–13,989, doi:10.1002/2014JD022453, 2014.
* * *
Acknowledgements added.

Pg 19, lines 22-31: This work is supported by the project GEO-CRADLE (Coordinating and integRating state-of-the-art Earth Observation Activities in the regions of North Africa, Middle East, and Balkans and Developing Links with GEO related initiatives towards GEOSS), Grant Agreement No. 690133, funded under European Union Horizon 2020 Programme - Topic: SC5-18b-2015, Integrating North African, Middle East and Balkan Earth Observation capacities in GEOSS. We would like also to acknowledge the support for international research staff exchange by REQUA (Regional climate-air quality interactions) project (FP7-PEOPLE-2013-IRSES - Marie Curie Action, PIRSES -GA -2013 -612671) and ACTRIS-2 project (Grand Agreement No. 654109, funded under European Union's Horizon 2020 programme). CALIPSO data were provided by NASA. LIVAS team thanks the

ICARE Data and Services Center (http://www.icare.univ-lille1.fr/) for providing access to CALIPSO data used for the production of LIVAS dataset.

Response to **Anonymous Referee #3** concerning the paper *"Dust size parameterization in RegCM4: Impact on aerosol burden and radiative forcing"* ([http://www.atmos-chem-phys-discuss.net/acp-2016-434](http://www.atmos-chem-phys-discuss.net/acp-2016-434)).

November 30, 2016
Dear editorial and respected reviewer,

Thank you very much for reviewing our manuscript and providing us with such a constructive feedback. We believe that your comments helped us highlight some critical aspects of the paper and add important content which elevates the quality of our work. All the short comments and typographical errors were corrected in the new version of the paper, while comments that needed more explanation and additional material are discussed below. The structure of the responses includes (1) comments from Referee, (2) a detailed response (3) changes implemented in the new version of the paper. Quotes from the initial version of the paper are highlighted with pale red along with the corresponding page and lines. Quotes from the new version of the paper are highlighted with pale blue along with the corresponding page and lines. The specific phrases changed-modified are underlined for convenience. In cases where the quotes are discussed but not changed are highlighted with grey. I sincerely hope you will be fully satisfied with all the changes we have made.

Best Regards,
Athanasios Tsikerdekis
* * *
1) I suggest to join all three datasets in the validation part. It makes the reader easier to see the difference between the two aerosol size bin discretization schemes in the view of the reference observational data. Of course, this holds only for those quantities, where observations are available.

We agree with the reviewer and therefore Figure 12 was updated and now includes the LIVAS DOD annual cycles along with the DUST4 and DUST12 experiments. Future readers can now directly compare DOD of LIVAS with the two experiments in the dust size bin discretization Section of the paper "3.2 Comparison of 4-bin and 12-bin experiments". Although since the DUST12 experiment is not discussed in the Section "3.1 Evaluation", we believe it would be confusing to include it in prior plots. Also, the following phrase was updated:
P 14, L 27: Figure 12 shows the annual cycle of DOD for the DUST4 and DUST12 experiments.
P 16, L 1: Figure 12 shows the annual cycle of DOD for LIVAS, DUST4 and DUST12 experiments.

[Figure]

**Figure 12. Dust optical depth annual cycle of LIVAS, DUST4 and DUST12 experiments for the period December 2006 to November 2014.**

2) The presented changes caused by the introduction of new aerosol size bin scheme are rather small. I agree with the other reviewer, that this requires a statistical significance test of the differences.

Thank you for highlighting the statistical significance issue. We have calculated the statistical significance of the differences using the monthly data for each grid point for the DOD, dust column burden and radiative forcing using the two-tailed paired t-test. In all variables and almost in all grid points the differences are statistically significant at the 95% confidence level (p.value < 0.05). We have also updated the text in the new version of the paper to highlight this issue. The plots shown below were not updated in the paper since we believe that they do not offer additional information to the reader; the vast majority of the grid points show differences that are statistically significant (shaded areas in the differences), hence we are just mentioning this in the text.

Pg 14, lines 13-20: The DOD percent increase is between 10.4% and 13% for all the subregions. Furthermore, there is a distinctive increase by 0.04 with the 12-bin model over the Sahara desert and especially along the Sahel region where the DOD values are higher (Figure 11c). In comparison with the DUST4 simulation the DUST12 simulation increases the deposition lifetime (column burden/total deposition flux) by 3.5 hours and 2 minutes for fine and coarse particles respectively. Consequently, that increases the dust column burden of fine (+4%) and coarse (+3%) particles (Figure 11f,i). The changes in the fine particles correlates better with the changes in DOD, because dust extinction coefficient is much higher for fine particles (<2.5μm) (Figure 1). Over the Middle East and the northern

part of the Arabian Peninsula we observe a distinct increase on the coarse dust column burden by 10mg•m-2.

Pg 15, lines 17-26: The DOD percent increase is between 10.4% and 13% for all the subregions. Furthermore, there is a distinctive increase by 0.04 with the 12-bin model over the Sahara desert and especially along the Sahel region where the DOD values are higher (Figure 11c). In comparison with the DUST4 simulation the DUST12 simulation increases the deposition lifetime (column burden/total deposition flux) by 3.5 hours and 2 minutes for fine and coarse particles respectively. Consequently, that increases the dust column burden of fine (+4%) and coarse (+3%) particles (Figure 11f,i). The changes in the fine particles correlates better with the changes in DOD, because dust extinction coefficient is much higher for fine particles (<2.5µm) (Figure 1). Over the Middle East and the northern part of the Arabian Peninsula we observe a distinct increase on the coarse dust column burden by 10mg•m-2. The differences of DOD and column burden between the two experiment, calculated from the monthly data for each grid, are statistically significant at the 95% confidence level according to a two-tailed paired t.test for almost all the grid points of the simulated domain.

Pg 16, lines 11-13: Similarly, at the surface the positive radiative forcing rise by 0.08W·m-2 (3.0%) and 0.09W·m-2 (2.7%) in eastern and western Sahara and 0.9W·m-2 (6.3%), 0.08W·m-2 (4.8%) and 0.9W·m-2 (6.5%) in western, central and eastern Mediterranean (Figure 14f).

Pg 17, lines 16-20: Similarly, at the surface the positive radiative forcing rise by 0.08W·m-2 (3.0%) and 0.09W·m-2 (2.7%) in eastern and western Sahara and 0.9W·m-2 (6.3%), 0.08W·m-2 (4.8%) and 0.9W·m-2 (6.5%) in western, central and eastern Mediterranean (Figure 14f). The shortwave and longwave radiative forcing differences between the two experiment, calculated from the monthly data for each grid, are statistically significant at the 95% confidence level according to a two-tailed paired t.test for almost all the grid points of the simulated domain.

[Figure]

[Figure]

3) Pg 3, lines 15-16: This statement is true, however please add 1-2 sentences on how this number affects the modeling of aerosols, at least in theory.

Indeed, adding a discussion at this point makes the initial statement more robust. Therefore, we have explained briefly why this is the case as it was suggested.

Pg 3, lines 15-16: An important component that affects the transport and the radiative properties of dust in climate modelling is the number of transport dust size bins.

Pg 3, lines 15-20: An important component that affects the transport and the radiative properties of dust in climate modelling is the number of transport dust size bins. Small dust particles, due to their weight, can travel over long distances and can efficiently reflect/backscatter the incoming shortwave solar radiation, while larger particles, with shorter atmospheric life, can effectively absorb and re-emit in the longwave spectrum. Thus, both the partitioning and the number of dust transport bins, used in atmospheric models, should carefully distinguish dust particles with contrasting radiative properties and transport characteristics.
* * *
4) Pg 5, lines 25-26: So aerosols cannot move from one bin to another meaning that there is no aerosol fragmentation or aerosol coagulation?

Yes. The current state of the model assumes that dust particles retain their size and they do not fragment into smaller particles during transport. Thus remaining in the same bin throughout their atmospheric life.

Pg 5, lines 25-26: Each transported bin is considered as a distinct tracer, which assumes that there is no mixing between the dust size bins.
* * *
5) Pg 6, lines 5-7: I would be interesting to compare the presented optical properties with those obtained by using only one effective diameter – i.e. the one that describes the bin (the center of it).

Thank you for giving us the opportunity to discuss a more technical aspect of our work. The bin specific extinction coefficient is displayed below for 4 and 12 bins (left and right respectively). The 1st method uses as effective particle radius the mean diameter of each size bin in order to calculate the extinction coefficient, while the 2nd method calculates the extinction coefficient for multiple radii within the range of each size bin and average them in the end. The 2nd method is numerically more accurate and thus it was adopted in our paper. The highest differences between the two methods are observed in fine particles (<2.5µm) where the optical properties change rapidly with particle size.

These informations could be valuable for future reader we have added the following plot as supplementary material and cited it also in the main text:

Pg 6, lines 4-7: The differences for all the optical parameters are relatively small, because the calculations were performed for multiple effective particle radii within the range of each size bin and averaged in the end, instead of using the mean effective radius of each size bin. Using this method the optical properties between the two experiments are almost identical.

Pg 6, lines 10-13: The differences for all the optical parameters are relatively small, because the calculations were performed for multiple effective particle radii within the range of each size bin and averaged in the end, instead of using the mean effective radius of each size bin (FigureS 1). Using this method the optical properties between the two experiments are almost identical.

[Figure]

*FigureS 1: Dust bin specific coefficient for 4 and 12 dust size bins. The first 1ˢᵗ method uses as effective particle radius the mean diameter of each size bin in order to calculate the extinction coefficient, while the 2ⁿᵈ method calculates the extinction coefficient for multiple radii within the range of each size bin and average them in the end.*
* * *
6) Pg 17, line 2: ..on emissions? I missed something in the manuscript? At this point, I would recommend to present an emission figure to gain some idea about its spatial distribution.

Thank you for pointing this out. Emission is incorrectly referred at this point. The two experiments have the same emission fluxes, since we are testing only the binning effect on DOD, dust column burden and RF. Therefore, we have removed emission and added DOD in the sentence.
Pg 17, Line 1-2: In the present study, we investigate the role of the modelled particle size distribution on the emission, total column burden and radiative effects of dust in a regional climate model.
Pg 18, Line 5-6: In the present study, we investigate the role of the modelled particle size distribution on DOD, dust column burden and radiative forcing of dust in a regional climate model.

Although since the spatial distribution of emission and deposition of dust might be interesting for future readers we have included the following plot as supplementary material and cite it in the main paper:
Pg 12, lines 4-5: The spatial distribution of wet deposition, dry deposition and surface emission fluxes is depicted for the fine and coarse particles in FigureS 3.

[Figure]

**FigureS 3: Wet deposition, dry deposition and surface emission fluxes of fine (a, b, c) and coarse (d, e, f) dust particles in DUST4 experiment for the period December 2006 to November 2014.**
* * *
7) Pg 18, line 14: The use of word 'underestimates' is not correct, as we have no comparison of RF figures with observation: there is no information on the reference values of the RF, so it can possibly be that the 4 bin approach is closer to the reality.

As correctly indicated the word "underestimates" may be misleading for the reader, therefore we have rephrased the sentence in order to be more comprehensive.

[revised manuscript text omitted]
"* ([http://www.atmos-chem-phys-discuss.net/acp-2016-434](http://www.atmos-chem-phys-discuss.net/acp-2016-434)).

December 21, 2016
Dear editorial and respected reviewer,

Thank you very much for your additional comments, suggestions and for recommending publication after minor revisions. The structure of the responses follow the same pattern as our previous response and it includes (1) comments from Referee, (2) a detailed response (3) changes implemented in the new version of the paper. Quotes from the initial version of the paper are highlighted with pale red along with the corresponding page and lines. Quotes from the new version of the paper are highlighted with pale blue along with the corresponding page and lines. The specific phrases changed-modified are underlined for convenience. In cases where the quotes are discussed but not changed are highlighted with grey.

Best Regards,
Athanasios Tsikerdekis
* * *
**1. The current title is significantly unclear and needs more work. The key subject of this paper is investigating the impact dust size parameterization in RegCM4 between two bin models, but the current title does not reflect it enough. I would strongly suggest to revise the title to make it clear. For example, "Impact of Dust size parameterizations in RegCM4 model on aerosol burden and radiative forcing".**

Thank you for your suggestion. We agree that the title may need some changes to make it clearer. Therefore, we have rephrased the title of the paper as "Impact of dust size parameterizations on aerosol burden and radiative forcing in RegCM4".
* * *
**2. The model horizontal and vertical grid resolution looks same as Zakey et al. (2006). I understand that some models have a limited choice for model configurations or computational cost could limit their choice too. But 18 layers are considered coarse because the region is highly affected by complex convective systems and boundary layer meteorology as well as synoptic scale meteorology. Please state why 18 layers are chosen and what are the uncertainty. Please add top layer height. Is it 90 mb?**

The 18 vertical layers is the default option for RegCM4 and it has been used in numerous recent RegCM4 studies (Akritidis et al., 2013; Alexandri et al., 2015; Giorgi et al., 2012; Nabat et al., 2012; Shalaby et al., 2012; Zanis et al., 2012) as it is also computationally a good compromise for long-term regional climate simulations. Nevertheless, there is an option to use 23 vertical layers (Güttler et al., 2013; O'Brien et al., 2012) in RegCM4 which we are planning to use in an upcoming study that will focus on the dust vertical profile and dust long-range transport. More vertical layers may help us improve the discrepancies that we have discussed in this manuscript (quote below), but more research is needed on that direction to reach a safe conclusion. Furthermore, the top of the atmosphere is set at 50 mbar in our simulations and we have included it in Table 1.

**Page 15, L5-L7:** Discrepancies in the vertical distribution of dust can also be misinterpreted from

possible local emission errors, the lack of simulated vertical levels or wet deposition biases. Further research is needed in this regard to reduce the mean vertical distribution bias of dust in the model.

| Top Layer Pressure | 50hPa |
|---|---|

**3. Please state what is additional computational burden for 12-bin scheme compared to the 4-bin scheme.**

Thank you for highlighting this, indeed it is a valuable addition to the paper and might be useful for future readers. Although we have to note that the computational cost may vary if we change the physic options of the model and domain size. The DUST4 experiment took 289716 seconds to completion which is about 3.35 days and DUST12 experiment took 440401 seconds which is approximately 5.01 days. Thus, DUST12 experiment increased the simulation cost by 66.8% in comparison to DUST4. We have added the following phrase in the manuscript:

**Page 10, L11-L13:** The DUST12 experiment increased the computational cost by 66.8% in comparison to DUST4. Although we have to note that the computational cost difference between the two experiments may vary under other model physics options and domain size.

**4. Page 19, L9, L12: Please add percent change after absolute values.**

We have added the percentages differences for the longwave RF.

**Page 19, L9-L13:** The negative changes in the surface radiative forcing are concentrated in the western part of Africa and extend over the Eastern Atlantic, where the near-constant annual dust plume is located. On the longwave spectrum DUST12 experiment enhances the positive RF at the surface and TOA by 0.1W·m-2 in a large portion of the Sahara desert, the northern part of the Arabian Peninsula and the middle east.

**Page 19, L9-L13:** The negative changes in the surface radiative forcing are concentrated in the western part of Africa and extend over the Eastern Atlantic, where the near-constant annual dust plume is located. On the longwave spectrum DUST12 experiment enhances the positive RF at the surface and TOA by 0.1W·m-2 (3% and 7% respectively) in a large portion of the Sahara desert, the northern part of the Arabian Peninsula and the middle east.

**5. The new section 3 title of "Results and Discussion" is unusual to me.**

The title of Section 3 has been renamed again into "Results".